# Contrastive CFG: Guiding Diffusion Sampling by Contrasting Positive and Negative Concepts

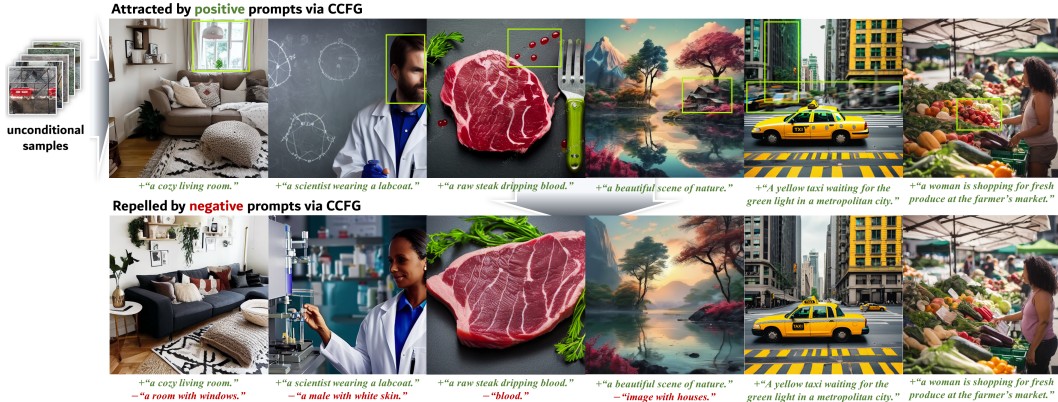

Figure 1: **Image samples generated from StableDiffusion 1.5 and SDXL by our proposed Contrastive CFG(CCFG) guidance term, using *positive* prompts (written in green) and *negative* prompts (written in red)**. The samples in the same column were from the same initial noise.

## Abstract

As Classifier-Free Guidance (CFG) has proven effective in conditional diffusion model sampling for improved condition alignment, many applications use a negated CFG term as a Negative Prompting (NP) to filter out unwanted features from samples. However, simply negating CFG guidance creates an inverted probability distribution, often distorting samples away from the marginal distribution. Inspired by recent advances in conditional diffusion models for inverse problems, here we present a novel method to achieve guidance toward the given condition using contrastive loss. Specifically, our guidance term aligns or repels the denoising direction based on the given condition through contrastive loss, achieving a similar guiding effect to traditional CFG for positive conditions while overcoming the limitations of existing negative guidance methods. Experimental results demonstrate that our approach effectively injects or removes the given concepts while maintaining sample quality across diverse scenarios, from simple class conditions to complex and overlapping text prompts.

## 1 Introduction

Classifier-Free Guidance (CFG) (Ho & Salimans, 2022) forms the key basis of modern text-guided generation with diffusion models. From Bayes rule, CFG constructs a Bayesian classifier $\nabla_{\boldsymbol{x}} \log p(\boldsymbol{c}|\boldsymbol{x}) = \nabla_{\boldsymbol{x}} \log p(\boldsymbol{x}|\boldsymbol{c}) - \nabla_{\boldsymbol{x}} \log p(\boldsymbol{x})$ without training additional external classifiers (Dhariwal & Nichol, 2021). In practice, it is common to emphasize the classifier vector direction with some constant $\gamma$, which corresponds to sharpening the posterior, i.e. $p(\boldsymbol{x})p(\boldsymbol{c}|\boldsymbol{x})^{\gamma}$. While this may potentially distort the sampling distribution (Du et al., 2023), it is known that the sample quality along with the text alignment increases.

While CFG was originally devised for *adhering* to the target condition, more often than not, there are cases where it is desirable to *avoid* sampling from some conditions. Canonical examples include

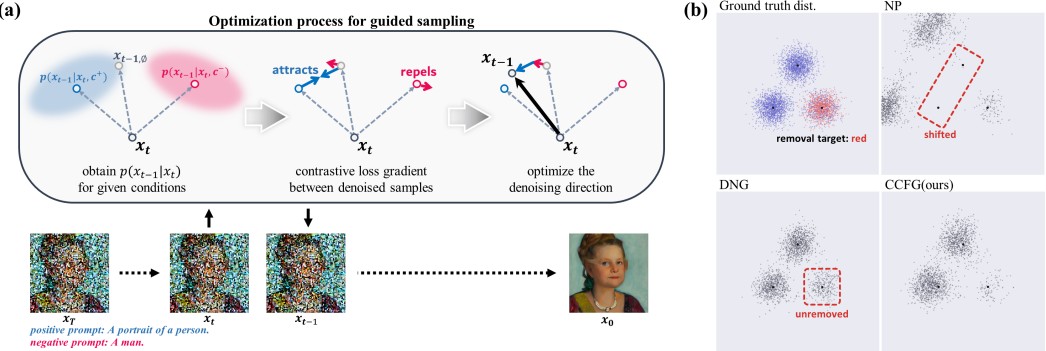

Figure 2: **Overview of the proposed guided sampling of CCFG.** (a) We pose guided sampling as an optimization problem that minimizes the contrastive loss of the positive and negative prompts, which has no computational overhead, yet avoids pitfalls of previous strategies such as NP. (b) Output distribution with different negative sampling methods on a toy dataset with two classes.

conditions that describe the poor quality of the image, or conditions that are related to harmful content (Gandikota et al., 2023; Wu et al., 2024). Often referred to as *Negative Prompting (NP)*, most implementations simply negate the vector direction of CFG, corresponding to inversely weighting with the classifier probability, i.e. $p(\boldsymbol{x})/p(\boldsymbol{c}|\boldsymbol{x})^\gamma$. Nonetheless, such näive implementation often leads to a decrease in the sample quality (Kim et al., 2025a; Armandpour et al., 2023). Specifically, due to the unboundness of an inverted probability distribution, sampling from $p(\boldsymbol{x})/p(\boldsymbol{c}|\boldsymbol{x})^\gamma$ leads to sampling from the low-density region or completely outside of the original density support.

To mitigate these drawbacks, by leveraging the recent advances in guided sampling methods (Chung et al., 2023; Kim et al., 2025b; Chung et al., 2024b; Yu et al., 2023; Ye et al., 2024), we reformulate the conditional guidance as an optimization problem with a positive or negative prompt-conditioned contrastive loss (Gutmann & Hyvärinen, 2010; Chen et al., 2020). Then, we derive a reverse diffusion sampling strategy with the optimized denoising direction. This results in a simple modification of CFG to the sampling process with little computational overhead. The resulting process, termed *Contrastive CFG (CCFG)*, optimizes the denoising direction by attracting or repelling the denoising direction for the given condition. Furthermore, the attracting and repelling forces are automatically controlled along the sampling process. Through extensive experiments, we verify that CCFG not only guides the samples to satisfy the wanted conditions, but also successfully avoid undesirable concepts while preserving the sample quality.

## 2 RELATED WORKS

### 2.1 CLASSIFIER-FREE GUIDANCE FOR DIFFUSION MODELS.

Diffusion models (Song et al., 2021b; Ho et al., 2020) are a class of generative models that learn the score function (Hyvärinen & Dayan, 2005) of the data distribution, and use this score function to reverse the forward noising process. The forward process, denoted with the time index $t$, is governed by a Gaussian kernel that the underlying data distribution $p(\boldsymbol{x}_0) \equiv p(\boldsymbol{x})$ eventually approximates the standard normal distribution at time $t = T$, i.e. $p(\boldsymbol{x}_T) \approx \mathcal{N}(0, \boldsymbol{I})$. The variance preserving forward transition kernel (Ho et al., 2020) is given as $p(\boldsymbol{x}_t|\boldsymbol{x}_0) = \mathcal{N}(\boldsymbol{x}_t; \sqrt{\bar{\alpha}_t}\boldsymbol{x}_0, (1-\bar{\alpha}_t)\boldsymbol{I})$. The reverse generative process follows a stochastic differential equation (SDE) (Song et al., 2021b) governed by the score function $\nabla_{\boldsymbol{x}_t} \log p(\boldsymbol{x}_t)$. To estimate this score function, one typically uses epsilon matching (Ho et al., 2020)

$$\theta^* = \arg\min_\theta \mathbb{E}\left[\|\boldsymbol{\epsilon}_\theta(\sqrt{\bar{\alpha}_t}\boldsymbol{x}_0 + \sqrt{1-\bar{\alpha}_t}\boldsymbol{\epsilon}) - \boldsymbol{\epsilon}\|_2^2\right], \tag{1}$$

which can be shown to be equivalent to denoising score matching (Vincent, 2011), $\boldsymbol{s}_\theta(\boldsymbol{x}_t) = \nabla_{\boldsymbol{x}_t} \log p(\boldsymbol{x}_t) = -\frac{1}{\sqrt{1-\bar{\alpha}_t}}\boldsymbol{\epsilon}_\theta(\boldsymbol{x}_t)$. By Tweedie's formula (Efron, 2011), one can recover the posterior mean $\hat{\boldsymbol{x}}(\boldsymbol{x}_t) = \mathbb{E}[\boldsymbol{x}_0|\boldsymbol{x}_t] = \frac{1}{\sqrt{\bar{\alpha}_t}}(\boldsymbol{x}_t - \sqrt{1-\bar{\alpha}_t}\boldsymbol{\epsilon}_\theta(\boldsymbol{x}_t))$. Moreover, it is common practice to train a *conditional* score function conditioned on the text prompt (Ho & Salimans, 2022) with random dropping to use it flexibly, either as $\boldsymbol{\epsilon}_\theta(\boldsymbol{x}_t, \boldsymbol{c})$ or $\boldsymbol{\epsilon}_\theta(\boldsymbol{x}_t) := \boldsymbol{\epsilon}_\theta(\boldsymbol{x}_t, \varnothing)$, where $\varnothing$ refers to the

null condition. In practice, a popular way of generating images through reverse sampling is through DDIM sampling (Song et al., 2021a), where a single iteration can be written as

$$\hat{\boldsymbol{x}}_\varnothing(\boldsymbol{x}_t) = (\boldsymbol{x}_t - \sqrt{1 - \bar{\alpha}_t}\boldsymbol{\epsilon}_\theta(\boldsymbol{x}_t, \varnothing))/\sqrt{\bar{\alpha}_t} \tag{2}$$

$$\boldsymbol{x}_{t-1} = \sqrt{\bar{\alpha}_{t-1}}\hat{\boldsymbol{x}}_\varnothing(\boldsymbol{x}_t) + \sqrt{1 - \bar{\alpha}_{t-1}}\boldsymbol{\epsilon}_\theta(\boldsymbol{x}_t, \varnothing), \tag{3}$$

where we defined $\hat{\boldsymbol{x}}_\varnothing(\boldsymbol{x}_t)$ as the posterior mean without the conditioning. Iterating (2) and (3) amounts to sampling from $p(\boldsymbol{x})$.

Plugging the wanted condition $\boldsymbol{c}^+$ into the conditional epsilon $\boldsymbol{\epsilon}_\theta(\boldsymbol{x}_t, \boldsymbol{c}^+)$ to sample from the conditional distribution $p(\boldsymbol{x}|\boldsymbol{c}^+)$ does not work well in practice, due to the guidance effect being too weak. To mitigate this downside, it is standard to use classifier-free guidance (CFG) (Ho & Salimans, 2022) at sampling time. The key idea is to use

$$\hat{\boldsymbol{\epsilon}}_{\boldsymbol{c}^+}^\gamma(\boldsymbol{x}_t) := \hat{\boldsymbol{\epsilon}}_\varnothing(\boldsymbol{x}_t) + \gamma(\hat{\boldsymbol{\epsilon}}_{\boldsymbol{c}^+}(\boldsymbol{x}_t) - \hat{\boldsymbol{\epsilon}}_\varnothing(\boldsymbol{x}_t)), \tag{4}$$

where we defined $\hat{\boldsymbol{\epsilon}}_{\boldsymbol{c}}(\boldsymbol{x}_t) := \boldsymbol{\epsilon}_\theta(\boldsymbol{x}_t, \boldsymbol{c})$[1]. Running DDIM sampling with $\hat{\boldsymbol{\epsilon}}_{\boldsymbol{c}^+}^\gamma(\boldsymbol{x}_t)$ in the place of $\hat{\boldsymbol{\epsilon}}_\varnothing(\boldsymbol{x}_t)$ leads to sampling from the gamma-powered distribution $p^\gamma(\boldsymbol{x}|\boldsymbol{c}^+) \propto p(\boldsymbol{x})p(\boldsymbol{c}^+|\boldsymbol{x})^\gamma$, a sharpened posterior. In this way, adherence to the condition $\boldsymbol{c}^+$ is emphasized.

## 2.2 GUIDED SAMPLING METHODS FOR DIFFUSION MODELS

Another popular way of posterior sampling with diffusion models is to use guided sampling, where the noised sample $\boldsymbol{x}_t$(Chung et al., 2023; Yu et al., 2023; Song et al., 2023; Ye et al., 2024) or its posterior mean $\hat{\boldsymbol{x}}_\varnothing(\boldsymbol{x}_t)$(Chung et al., 2024b; He et al., 2023) is added by appropriate guidance terms according to the gradient of a pre-defined energy function $\ell(\cdot)$ to be minimized. For instance, Decomposed Diffusion Sampling (Chung et al., 2024b) has the form of

$$\boldsymbol{x}_{t-1} = \sqrt{\bar{\alpha}_{t-1}}(\hat{\boldsymbol{x}}_\varnothing - \omega_t \nabla_{\hat{\boldsymbol{x}}_\varnothing}\ell(\hat{\boldsymbol{x}}_\varnothing)) + \sqrt{1 - \bar{\alpha}_{t-1}}\hat{\boldsymbol{\epsilon}}_\varnothing, \tag{5}$$

where $\omega_t$ is the step size, to generate sample $\boldsymbol{x}_0$ with small $\ell(\boldsymbol{x}_0)$. Chung et al. (2024a) presented that CFG can also be seen as such guided sampling, when we replace $\ell(\hat{\boldsymbol{x}}_\varnothing)$ in (5) with the following loss function similar to Score Distillation Sampling (SDS) (Poole et al., 2023).

$$\ell_{SDS}(\boldsymbol{x}) := \|\boldsymbol{\epsilon}_\theta(\sqrt{\bar{\alpha}_t}\boldsymbol{x} + \sqrt{1 - \bar{\alpha}_t}\boldsymbol{\epsilon}, \boldsymbol{c}) - \boldsymbol{\epsilon}\|_2^2$$

$$= \frac{\bar{\alpha}_t}{1 - \bar{\alpha}_t}\|\hat{\boldsymbol{x}}_{\boldsymbol{c}} - \boldsymbol{x}\|_2^2 \tag{6}$$

This leads to the alternative version of CFG called CFG++(Chung et al., 2024a) with $\gamma := \frac{2\bar{\alpha}_t}{1 - \bar{\alpha}_t}\omega_t$,

$$\boldsymbol{x}_{t-1} = \sqrt{\bar{\alpha}_{t-1}}(\hat{\boldsymbol{x}}_\varnothing + \gamma(\hat{\boldsymbol{x}}_{\boldsymbol{c}} - \hat{\boldsymbol{x}}_\varnothing)) + \sqrt{1 - \bar{\alpha}_{t-1}}\hat{\boldsymbol{\epsilon}}_\varnothing, \tag{7}$$

which is shown as a weight-rescaled version of CFG where the improvement is especially dominating at the early stage of reverse sampling.

## 2.3 NEGATIVE GUIDANCE IN DIFFUSION MODEL SAMPLING

In contrast to CFG that samples toward the $\boldsymbol{c}^+$, *negative* guidance aims to avoid samples that meet the unwanted condition $\boldsymbol{c}^-$. Negative guidance can be utilized as itself when samples from a broad unconditional distribution, yet excluding certain unwanted features, are desired. When the unconditional distribution is too broad to be meaningful(*e.g.*, text-to-image models), negative guidance is often combined with the positive guidance of CFG.

**Inversely proportional distribution-based methods.** The simplest case (Gandikota et al., 2023; Koulischer et al., 2025; Armandpour et al., 2023; Liu et al., 2022), often called *negative prompting*(NP), negates the guidance direction of CFG in (4) such as

$$\hat{\boldsymbol{\epsilon}}_{\boldsymbol{c}^-}^\gamma(\boldsymbol{x}_t) := \hat{\boldsymbol{\epsilon}}_\varnothing(\boldsymbol{x}_t) - \gamma(\hat{\boldsymbol{\epsilon}}_{\boldsymbol{c}^-}(\boldsymbol{x}_t) - \hat{\boldsymbol{\epsilon}}_\varnothing(\boldsymbol{x}_t)), \tag{8}$$

---

[1]With a slight abuse of notation, we omit the dependence on $\boldsymbol{x}_t$ when it is clear from context.

where the goal is to avoid sampling from $\boldsymbol{c}^-$. Note that this corresponds to sampling from $p^{-\gamma}(\boldsymbol{x}|\boldsymbol{c}^-) := p(\boldsymbol{x})/p(\boldsymbol{c}^-|\boldsymbol{x})^\gamma$, a joint distribution inversely proportional to the posterior likelihood. When the goal is to sample from $\boldsymbol{c}^+$ *while* avoiding $\boldsymbol{c}^-$, one uses (Ban et al., 2024)

$$\hat{\boldsymbol{\epsilon}}^\gamma_{\boldsymbol{c}^+,\boldsymbol{c}^-} := \hat{\boldsymbol{\epsilon}}_\varnothing + \gamma(\hat{\boldsymbol{\epsilon}}_{\boldsymbol{c}^+} - \hat{\boldsymbol{\epsilon}}_\varnothing) - \gamma(\hat{\boldsymbol{\epsilon}}_{\boldsymbol{c}^-} - \hat{\boldsymbol{\epsilon}}_\varnothing)$$
$$= \hat{\boldsymbol{\epsilon}}_\varnothing + \gamma(\hat{\boldsymbol{\epsilon}}_{\boldsymbol{c}^+} - \hat{\boldsymbol{\epsilon}}_{\boldsymbol{c}^-}) \tag{9}$$

In both cases, pushing away from $\boldsymbol{c}^-$ is governed by the *negation* of the vector direction $(\hat{\boldsymbol{\epsilon}}_{\boldsymbol{c}^-} - \hat{\boldsymbol{\epsilon}}_\varnothing)$.

Several works proposed empirical improvements for aggregating positive and negative concepts in (9) for the situation where the negative guidance term conflicts with the pre-given positive conditions. This includes the on-the-fly control of the negative guidance scale(Schramowski et al., 2023), or canceling the negative guidance component parallel to the CFG term(Armandpour et al., 2023).

**Utilizing complementary conditions.** Koulischer et al. (2025) proposed dynamic negative guidance (DNG) and proposed sampling from $p(\boldsymbol{x})p(\neg\boldsymbol{c}^-|\boldsymbol{x})^\gamma$, where $\neg\boldsymbol{c}^-$ is defined as a hypothetical condition such that $p(\neg\boldsymbol{c}^-|\boldsymbol{x}) = 1 - p(\boldsymbol{c}^-|\boldsymbol{x})$. This $\neg\boldsymbol{c}^-$ can also be seen as a union of all possible input conditions except $\boldsymbol{c}^-$, as a complementary set of $\boldsymbol{c}^-$. Applying the Bayes rule, one can show that

$$\nabla_{\boldsymbol{x}_t} \log p(\boldsymbol{x}_t|\neg\boldsymbol{c}^-) = \nabla_{\boldsymbol{x}_t} \log p(\boldsymbol{x}_t) - \gamma(\boldsymbol{x}_t, \boldsymbol{c}^-)(\nabla_{\boldsymbol{x}_t} \log p(\boldsymbol{x}_t|\boldsymbol{c}^-) - \nabla_{\boldsymbol{x}_t} \log p(\boldsymbol{x}_t)), \tag{10}$$

where $\gamma(\boldsymbol{x}_t, \boldsymbol{c}^-) := \frac{p(\boldsymbol{c}^-|\boldsymbol{x}_t)}{1-p(\boldsymbol{c}^-|\boldsymbol{x}_t)}$, which can be approximated during the reverse diffusion process to adjust the guidance scale dynamically.

## 3 CONTRASTIVE CLASSIFIER-FREE GUIDANCE

### 3.1 MOTIVATION

The downside of the NP term in (8) can be explained in two different aspects: the sampling distribution involves the inverted probability $p(\boldsymbol{c}^-|\boldsymbol{x})^{-\gamma}$, which quickly dominates over $p(\boldsymbol{x})$ as $\gamma$ grows (Koulischer et al., 2025) and heavily distorts the sampling distribution off the supports of the marginal data distribution. This can also be seen from the NP term $(\hat{\boldsymbol{\epsilon}}_{\boldsymbol{c}^-} - \hat{\boldsymbol{\epsilon}}_\varnothing)$ becoming bigger in regions far from $\boldsymbol{c}^-$, unnecessarily pushing further away. Figure 2-(b) visualizes this issue with a toy dataset consisting of two classes, where two modes contain the blue class and the other mode is mixed with blues and reds. When NP sampling was done with the red class, the output distribution avoided the red mode but the other two modes were entirely shifted from the original location. The ideal behavior of negative guidance is that the strength of the guidance term decreases to zero as $\boldsymbol{x}_t$ becomes irrelevant to the given condition, being equivalent to unconditional sampling.

Meanwhile, sampling from the complementary condition $\neg\boldsymbol{c}^-$ (Koulischer et al., 2025) resolved this issue by weighting the NP guidance term. However, the biggest limitation is that its sample faithfully avoids $\boldsymbol{c}^-$ only when $p(\boldsymbol{c}^-|\boldsymbol{x}_t) \approx 1$ for all $\boldsymbol{x}_t$ that satisfies $\boldsymbol{c}^-$. If the conditions are not mutually exclusive, the output distribution still includes the data that agrees with $\boldsymbol{c}^-$, as the samples are still retained in the third node as a blue class in Figure 2-(b). This limitation is especially crucial for text-to-image (T2I) models (*e.g.* StableDiffusion (Rombach et al., 2022)) which take a wide range of overlapping prompts, and therefore each prompt has low $p(\boldsymbol{c}|\boldsymbol{x})$.

### 3.2 DERIVATION OF CCFG

Similar to redefining positive CFG as a guided sampling that assimilates the denoising direction with the conditioned noise (Chung et al., 2024a; Kim et al., 2025b), we implement sampling that negates certain conditions with an appropriate objective function. Considering this task as optimizing the data in the sampling process closer to or further away from a given condition, we propose using contrastive loss (Hadsell et al., 2006; Chen et al., 2020) as the objective function for positive/negative prompt guidance. Contrastive loss assimilates features with the same context while distancing semantically unrelated features. Among various contrastive loss concepts, we found that Noise Contrastive Estimation (NCE) (Gutmann & Hyvärinen, 2010) best suits our situation.

Specifically, NCE parameterizes the data by performing logistic regression to discriminate the observed data from the noise data distribution. With a modeled data distribution $p_\theta(\boldsymbol{x})$ parameterized

by $\theta$ and pre-defined noise distribution $q(\boldsymbol{x})$, the logit of $\boldsymbol{x}$ for the logistic regression is defined as

$$l_\theta(\boldsymbol{x}) := \log p_\theta(\boldsymbol{x}) - \log q(\boldsymbol{x}), \tag{11}$$

which formulates the NCE loss as

$$\begin{aligned} \mathcal{L}_{NCE} &:= -y \log \sigma(l_\theta(x)) - (1 - y) \log(1 - \sigma(l_\theta(x))) \\ &= -y \log \frac{p_\theta(\boldsymbol{x})}{p_\theta(\boldsymbol{x}) + q(\boldsymbol{x})} - (1 - y) \log \frac{q(\boldsymbol{x})}{p_\theta(\boldsymbol{x}) + q(\boldsymbol{x})} \end{aligned} \tag{12}$$

where $y$ is 1 if $\boldsymbol{x}$ is the real data and 0 if $\boldsymbol{x}$ is sampled from $q(\boldsymbol{x})$. Although NCE mostly updates the model parameter with a given positive or noise data, we optimize a datapoint with a pre-trained model that nicely parameterizes $p_\theta(\boldsymbol{x})$.

When we guide the denoising of the noised data $\boldsymbol{x}_t$ to satisfy $\boldsymbol{c}$, we set $q(\boldsymbol{x}) := p(\boldsymbol{x}_{t-1}|\boldsymbol{x}_t, \varnothing)$ and parameterize $p_\theta(\boldsymbol{x})$ with the pre-trained diffusion model $p(\boldsymbol{x}_{t-1}|\boldsymbol{x}_t, \boldsymbol{c})$ to optimize $\boldsymbol{\epsilon}$ with the corresponding NCE loss:

$$\ell^+(\boldsymbol{\epsilon}) := -\log \frac{p_\theta(\hat{\boldsymbol{x}}_{t-1}(\boldsymbol{\epsilon}))}{p_\theta(\hat{\boldsymbol{x}}_{t-1}(\boldsymbol{\epsilon})) + q(\hat{\boldsymbol{x}}_{t-1}(\boldsymbol{\epsilon}))} \tag{13}$$

Here, $\hat{\boldsymbol{x}}_{t-1}(\boldsymbol{\epsilon})$ is a mean prediction of $\boldsymbol{x}_{t-1}$ from $\boldsymbol{x}_t$ with $\boldsymbol{\epsilon}$. DDIM (Song et al., 2021a) states a closed form representation of $p(\boldsymbol{x}_{t-1}|\boldsymbol{x}_t)$ with a given noise prediction $\boldsymbol{\epsilon}$ as a Gaussian distribution $\mathcal{N}(\boldsymbol{\mu}(\boldsymbol{x}_t, \boldsymbol{\epsilon}), \sigma_t^2)$ with a certain choice of variance $\sigma_t^2$ and

$$\boldsymbol{\mu}(\boldsymbol{x}_t, \boldsymbol{\epsilon}) = \frac{\sqrt{\bar{\alpha}_{t-1}}}{\sqrt{\bar{\alpha}_t}} \boldsymbol{x}_t - \left( \frac{\sqrt{\bar{\alpha}_{t-1}}\sqrt{1 - \bar{\alpha}_t}}{\sqrt{\bar{\alpha}_t}} - \sqrt{1 - \bar{\alpha}_{t-1} - \sigma_t^2} \right) \boldsymbol{\epsilon} \tag{14}$$

Therefore, $p(\boldsymbol{x}_{t-1}|\boldsymbol{x}_t, \varnothing) = \mathcal{N}(\boldsymbol{\mu}(\boldsymbol{x}_t, \hat{\boldsymbol{\epsilon}}_\varnothing), \sigma_t^2)$ and $p(\boldsymbol{x}_{t-1}|\boldsymbol{x}_t, \boldsymbol{c}) = \mathcal{N}(\boldsymbol{\mu}(\boldsymbol{x}_t, \hat{\boldsymbol{\epsilon}}_{\boldsymbol{c}}), \sigma_t^2)$. By replacing $\hat{\boldsymbol{x}}_{t-1}(\boldsymbol{\epsilon})$ with $\boldsymbol{\mu}(\boldsymbol{x}_t, \boldsymbol{\epsilon})$, the loss function in (13) can be simplified as

$$\ell^+(\boldsymbol{\epsilon}) = -\log \frac{e^{-\|\boldsymbol{\mu}(\boldsymbol{x}_t, \boldsymbol{\epsilon}) - \boldsymbol{\mu}(\boldsymbol{x}_t, \hat{\boldsymbol{\epsilon}}_{\boldsymbol{c}})\|_2^2 / 2\sigma_t^2}}{e^{-\|\boldsymbol{\mu}(\boldsymbol{x}_t, \boldsymbol{\epsilon}) - \boldsymbol{\mu}(\boldsymbol{x}_t, \hat{\boldsymbol{\epsilon}}_{\boldsymbol{c}})\|_2^2 / 2\sigma_t^2} + e^{-\|\boldsymbol{\mu}(\boldsymbol{x}_t, \boldsymbol{\epsilon}) - \boldsymbol{\mu}(\boldsymbol{x}_t, \hat{\boldsymbol{\epsilon}}_\varnothing)\|_2^2 / 2\sigma_t^2}} \tag{15}$$

$$= -\log \frac{e^{-\tau_t \|\boldsymbol{\epsilon} - \hat{\boldsymbol{\epsilon}}_{\boldsymbol{c}}\|_2^2}}{e^{-\tau_t \|\boldsymbol{\epsilon} - \hat{\boldsymbol{\epsilon}}_{\boldsymbol{c}}\|_2^2} + e^{-\tau_t \|\boldsymbol{\epsilon} - \hat{\boldsymbol{\epsilon}}_\varnothing\|_2^2}} \tag{16}$$

Here, $\tau_t$ is a coefficient that depends on the noise scheduling and the selection of $\sigma_t$, working as the "temperature" on the logit (Chen et al., 2020). Although $\tau_t$ can be defined as timestep-dependent based on the diffusion reverse process, we found that setting $\tau_t$ as a constant is sufficient to yield stable and preferable results, while being simple to implement. A more detailed analysis of the selection of $\tau_t$ can be found in Appendix B.3.

If we take the derivative of $\ell^+(\boldsymbol{\epsilon})$ with respect to $\boldsymbol{\epsilon}$ at $\boldsymbol{\epsilon} = \hat{\boldsymbol{\epsilon}}_\varnothing$ and update $\boldsymbol{\epsilon}$ with step size $\gamma_t$, we can obtain the positive guidance term:

$$\hat{\boldsymbol{\epsilon}}_{\boldsymbol{c}, \ell^+}^{\gamma_t, \tau_t} := \hat{\boldsymbol{\epsilon}}_\varnothing - \gamma_t \nabla_{\boldsymbol{\epsilon}} \ell^+(\hat{\boldsymbol{\epsilon}}_\varnothing) \tag{17}$$

$$= \hat{\boldsymbol{\epsilon}}_\varnothing + \frac{2\omega_t}{1 + e^{-\tau\|\hat{\boldsymbol{\epsilon}}_\varnothing - \hat{\boldsymbol{\epsilon}}_{\boldsymbol{c}}\|_2^2}} (\hat{\boldsymbol{\epsilon}}_{\boldsymbol{c}} - \hat{\boldsymbol{\epsilon}}_\varnothing) \tag{18}$$

Here, $\omega_t := \tau_t \gamma_t$ governs the overall guidance strength, which works similarly to the guidance scale in conventional CFG.

Similarly, to utilize the same objective function to avoid $\boldsymbol{c}$, we set $q(\boldsymbol{x})$ and $p_\theta(\boldsymbol{x})$ the same as above but treat $\boldsymbol{x}_t$ as a sample from $q(\boldsymbol{x})$. When we perform gradient descent on $\hat{\boldsymbol{\epsilon}}_\varnothing$ to minimize the following loss term, we get

$$\ell^-(\boldsymbol{\epsilon}) := -\log \frac{q(\hat{\boldsymbol{x}}_{t-1}(\boldsymbol{\epsilon}))}{p_\theta(\hat{\boldsymbol{x}}_{t-1}(\boldsymbol{\epsilon})) + q(\hat{\boldsymbol{x}}_{t-1}(\boldsymbol{\epsilon}))} \tag{19}$$

$$= -\log \frac{e^{-\tau_t \|\boldsymbol{\epsilon} - \hat{\boldsymbol{\epsilon}}_\varnothing\|_2^2}}{e^{-\tau_t \|\boldsymbol{\epsilon} - \hat{\boldsymbol{\epsilon}}_{\boldsymbol{c}}\|_2^2} + e^{-\tau_t \|\boldsymbol{\epsilon} - \hat{\boldsymbol{\epsilon}}_\varnothing\|_2^2}} \tag{20}$$

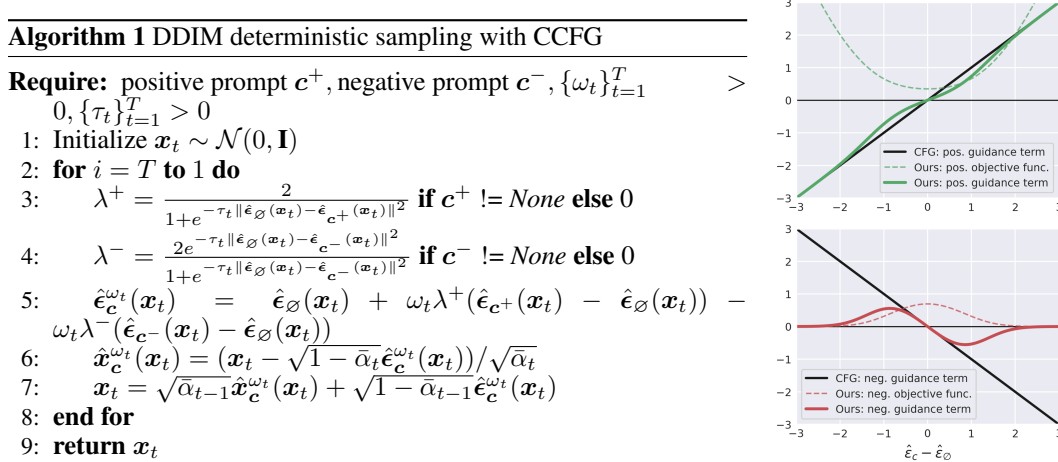

---

**Algorithm 1** DDIM deterministic sampling with CCFG

---

**Require:** positive prompt $\boldsymbol{c}^+$, negative prompt $\boldsymbol{c}^-$, $\{\omega_t\}_{t=1}^T$ $>$ $0, \{\tau_t\}_{t=1}^T > 0$
1: Initialize $\boldsymbol{x}_t \sim \mathcal{N}(0, \mathbf{I})$
2: **for** $i = T$ **to** $1$ **do**
3: $\quad \lambda^+ = \frac{2}{1+e^{-\tau_t \|\hat{\boldsymbol{\epsilon}}_\varnothing(\boldsymbol{x}_t) - \hat{\boldsymbol{\epsilon}}_{\boldsymbol{c}^+}(\boldsymbol{x}_t)\|^2}}$ **if** $\boldsymbol{c}^+$ != *None* **else** 0
4: $\quad \lambda^- = \frac{2e^{-\tau_t \|\hat{\boldsymbol{\epsilon}}_\varnothing(\boldsymbol{x}_t) - \hat{\boldsymbol{\epsilon}}_{\boldsymbol{c}^-}(\boldsymbol{x}_t)\|^2}}{1+e^{-\tau_t \|\hat{\boldsymbol{\epsilon}}_\varnothing(\boldsymbol{x}_t) - \hat{\boldsymbol{\epsilon}}_{\boldsymbol{c}^-}(\boldsymbol{x}_t)\|^2}}$ **if** $\boldsymbol{c}^-$ != *None* **else** 0
5: $\quad \hat{\boldsymbol{\epsilon}}_{\boldsymbol{c}}^{\omega_t}(\boldsymbol{x}_t) = \hat{\boldsymbol{\epsilon}}_\varnothing(\boldsymbol{x}_t) + \omega_t \lambda^+(\hat{\boldsymbol{\epsilon}}_{\boldsymbol{c}^+}(\boldsymbol{x}_t) - \hat{\boldsymbol{\epsilon}}_\varnothing(\boldsymbol{x}_t)) - \omega_t \lambda^-(\hat{\boldsymbol{\epsilon}}_{\boldsymbol{c}^-}(\boldsymbol{x}_t) - \hat{\boldsymbol{\epsilon}}_\varnothing(\boldsymbol{x}_t))$
6: $\quad \hat{\boldsymbol{x}}_{\boldsymbol{c}}^{\omega_t}(\boldsymbol{x}_t) = (\boldsymbol{x}_t - \sqrt{1 - \bar{\alpha}_t}\hat{\boldsymbol{\epsilon}}_{\boldsymbol{c}}^{\omega_t}(\boldsymbol{x}_t)) / \sqrt{\bar{\alpha}_t}$
7: $\quad \boldsymbol{x}_t = \sqrt{\bar{\alpha}_{t-1}}\hat{\boldsymbol{x}}_{\boldsymbol{c}}^{\omega_t}(\boldsymbol{x}_t) + \sqrt{1 - \bar{\alpha}_{t-1}}\hat{\boldsymbol{\epsilon}}_{\boldsymbol{c}}^{\omega_t}(\boldsymbol{x}_t)$
8: **end for**
9: **return** $\boldsymbol{x}_t$

---

Figure 3: Left: A detailed algorithm for diffusion model sampling with CCFG. Right: The plot of the utilized objective function and its guidance term, in positive (up) and negative guidance (down). For visual simplicity, we considered one-dimensional data and fixed $\tau_t$ to 1.0.

which results in the following update:

$$\hat{\boldsymbol{\epsilon}}_{\boldsymbol{c},\ell^-}^{\gamma_t,\tau_t} := \hat{\boldsymbol{\epsilon}}_\varnothing - \frac{2\omega_t e^{-\tau_t \|\hat{\boldsymbol{\epsilon}}_\varnothing - \hat{\boldsymbol{\epsilon}}_{\boldsymbol{c}}\|_2^2}}{1 + e^{-\tau_t \|\hat{\boldsymbol{\epsilon}}_\varnothing - \hat{\boldsymbol{\epsilon}}_{\boldsymbol{c}}\|_2^2}}(\hat{\boldsymbol{\epsilon}}_{\boldsymbol{c}} - \hat{\boldsymbol{\epsilon}}_\varnothing). \tag{21}$$

Explicit derivation of (18) and (21) can be found in Appendix B.2.

Algorithm 1 congregates the two scenarios and performs deterministic[2] DDIM sampling to satisfy or remove the given condition $\boldsymbol{c}$. Compared to the conventional CFG, an additional weighting coefficient was added to the guidance term of each step. We plot the objective functions and their guidance term of CCFG in the right side of Figure 3 as the difference between $\hat{\boldsymbol{\epsilon}}_\varnothing$ and $\hat{\boldsymbol{\epsilon}}_{\boldsymbol{c}}$ changes. As $\|\hat{\boldsymbol{\epsilon}}_{\boldsymbol{c}} - \hat{\boldsymbol{\epsilon}}_\varnothing\|$ increases, the positive guidance term approximates a linear function as the original CFG. On the other hand, the negative contrastive loss and its guidance term flatten out to 0 as $\|\hat{\boldsymbol{\epsilon}}_{\boldsymbol{c}} - \hat{\boldsymbol{\epsilon}}_\varnothing\|$ gets bigger, while the negative guidance of NP diverges. This demonstrates the instability of the NP term, and how CCFG can resolve this issue by canceling the gradient guidance when $\hat{\boldsymbol{\epsilon}}_\varnothing$ from the current sample $\boldsymbol{x}_t$ is sufficiently unrelated to the condition. The behavior of the objective function is also changed by the choice of $\tau_t$, whose effect on the changes of the guidance term is described in Appendix B.3.

Overall, we list the advantages of the proposed CCFG as follows: First, it faithfully removes unwanted concepts compared to complementary condition-based methods such as DNG. Concurrently, contrary to NP, it minimizes the influence on the sampling odds of unrelated data. Finally, the negative guidance of CCFG is a stand-alone method, while many empirical adjustments of NP require the presence of positive prompts to be applicable Armandpour et al. (2023); Schramowski et al. (2023).

## 4 EXPERIMENTAL RESULTS

We tested how CCFG steers the sample to satisfy or exclude certain conditions, along with its effect on the sample quality. We then thoroughly compared its performance to the following baselines: For scenarios where only one of the positive and negative conditions is used, we tested the conventional CFG, CFG++ (Chung et al., 2024a), NP, and DNG. In more complex scenarios involving both positive and negative conditions, we also tested Perp-Neg (Armandpour et al., 2023), SLD (Schramowski et al., 2023) and SAFREE (Yoon et al., 2024) as empirical improvements of NP in the presence of positive conditions. We conducted experiments across a range of scenarios where CFG is applicable, from simple class-conditioned image generation to text-to-image (T2I) generation.

---

[2]The selected variance $\sigma_t$ for $\tau_t$ is only used to obtain the guidance term, and it's not required to perform DDIM sampling in a non-deterministic manner with $\sigma_t$.

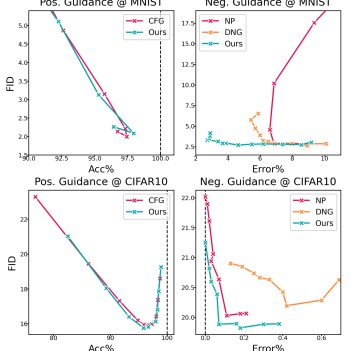

| Method | **ImageNet-1k** | | | | |
|--------|----------|----------|-------------|-----------|-----------|
| | Pos. Acc. | Neg. Acc. | Overall Acc. | Aesthetic | FD-DINOv2 |
| NP | 0.4959 | 0.9996 | 0.4957 | 4.006 | 456.78 |
| DNG | 0.8307 | 0.9886 | 0.8212 | 4.112 | 215.07 |
| Perp-Neg | 0.8544 | 0.9958 | 0.8508 | 4.209 | 214.93 |
| SLD | 0.7785 | 0.9945 | 0.7742 | 4.094 | 246.19 |
| CCFG | 0.9101 | 0.9932 | 0.9039 | 4.254 | 190.71 |

Figure 4: Left: The plot between the classification accuracy and FID with class-conditional and class-removal sampling, on MNIST and CIFAR10. Right: Quantitative results for positive and negative guidance on ImageNet-1k.

## 4.1 CLASS-CONDITIONED MODELS

**Positive-only and negative-only guidance.** To show a clear effect of concept negation and its trade-off between sample quality, we first employed MNIST (LeCun et al., 1998) and CIFAR10 (Krizhevsky et al., 2009), which comprise only 10 classes(*e.g.*, 10% chance of unwanted class in unconditional sampling) where each class is clearly distinctive from one another. We sampled 10,000 images with a positive or negative guidance for a randomly selected class, and calculated their classification accuracy to quantify the alignment to the given positive or negative class. The classification was done by a separate external classifier. Fréchet Inception Distance (FID) (Heusel et al., 2017) between the sampled images and the real data was measured to quantify the overall sampled image quality. See Appendix D.2 for more details of the experimental settings.

Figure 4 shows the curve of the classification rate against FID with varying guidance scales. For the positive guidance, the curve of CCFG almost lies on the curve of the original CFG, indicating that positive CCFG behaves similarly to the widely-used guidance term. The curve of the negative CCFG is located at the bottom left in both MNIST and CIFAR10, outperforming the NP and DNG. In contrast to NP, which corrupts the image quality and DNG, which fails to thoroughly remove the unwanted classes, CCFG achieve similar or better precision with higher sample quality.

**Guidance with positive and negative prompts.** Beyond datasets with categorically distinct classes (e.g., MNIST, CIFAR10), large-scale class-conditional settings such as ImageNet-1k (Deng et al., 2009) contain many fine-grained pairs whose features are strongly entangled (e.g., *malalmute* vs. *siberian husky*). This motivates evaluating guidance schemes for a task that (i) pull samples toward a desired class while (ii) repelling them from a closely related, potentially confounding class.

We curated 100 fine-grained class pairs with strong visual proximity $(c^+, c^-)$ from ImageNet-1k(See Appendix D.2 for the construction) and generated 100 images for each pair, applying positive guidance toward $c^+$ while applying negative guidance to $c^-$. We report (i) Positive accuracy: fraction of samples classified as $c^+$ by an external classifier; (ii) Negation ratio: fraction not classified as $c^-$; and (iii) Overall accuracy: the product of the two, simultaneously capturing the alignment and exclusion. To quantify perceptual quality independent of class labels, we report the mean Aesthetic Score (Schuhmann, 2022) over all generated samples. For reference against standard positive-only guidance, we also measure positive accuracy by sampling 100 random classes and generating 100 images per class with positive guidance only.

The table in Figure 4 summarizes the trade-off between class alignment, successful negation, and perceptual quality. CCFG attains the highest *overall accuracy* across pairs, indicating the most faithful target-class alignment while reliably excluding the forbidden class. NP, whose inverted weighting can over-repel, shows degradation of the positive accuracy and the perceptual quality. DNG, whose usage of complementary-conditions can under-repel when classes overlap, shows the lowest negation ratio. Perp-Neg, while demonstrating the highest negation ratio, suffers minimum positive accuracy compared to SLD, DNG and CCFG. Meanwhile, CCFG obtains the highest positive accuracy and near-perfect negation without sacrificing perceptual quality, demonstrating that CCFG's

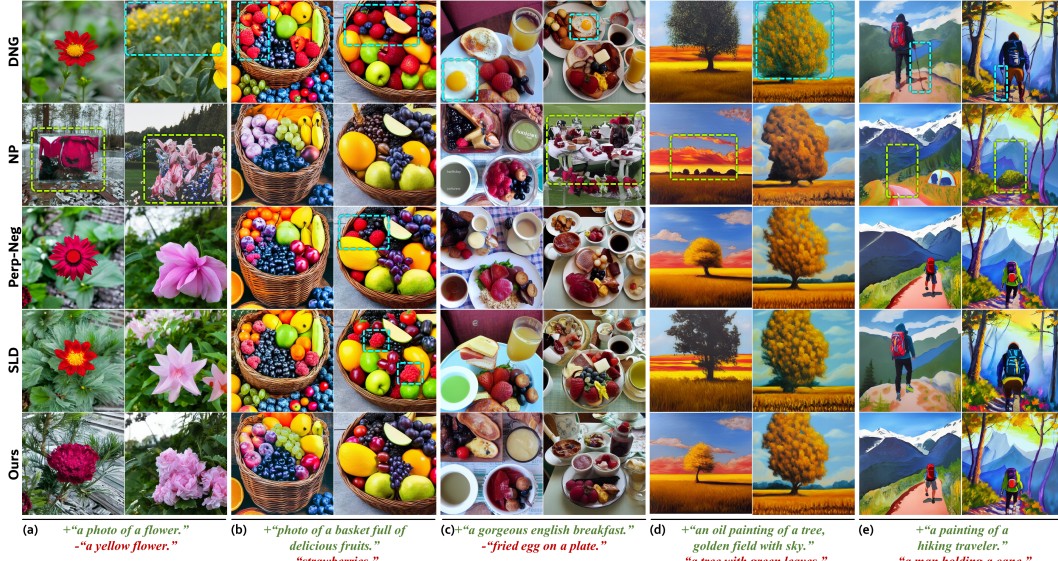

(a) +"a photo of a flower." −"a yellow flower."  (b) +"photo of a basket full of delicious fruits." −"strawberries."  (c)+"a gorgeous english breakfast." −"fried egg on a plate."  (d) +"an oil painting of a tree, golden field with sky." −"a tree with green leaves."  (e) +"a painting of a hiking traveler." −"a man holding a cane."

Figure 5: Image samples from StableDiffusion 1.5 using positive prompts (written in green) and negative prompts (written in red), across various negative sampling methods. The samples in the same column were sampled from the same initial noise. The regions where the negative prompt hasn't fully removed are highlighted in blue, and the regions where they failed to follow the positive prompt faithfully are highlighted in yellow.

| | **StableDiffusion 1.5** | | | | | | | | | | | | |
|---|---|---|---|---|---|---|---|---|---|---|---|---|---|
| | **GenEval[↑]** | | | | | | | **DPGBench[↑]** | | | | | |
| Method | single_obj. | two_obj. | count | colors | position | color_attr. | overall | attr. | entity | global | relation | other | overall |
| CFG | 0.959 | 0.301 | **0.388** | 0.723 | 0.028 | 0.045 | 0.407 | 0.734 | **0.731** | 0.839 | 0.794 | 0.556 | 0.635 |
| CFG++ | 0.991 | 0.434 | 0.290 | 0.739 | 0.047 | 0.055 | 0.426 | 0.722 | 0.720 | 0.769 | **0.810** | 0.552 | 0.630 |
| CCFG | **0.986** | **0.422** | 0.284 | **0.782** | 0.058 | **0.058** | **0.432** | **0.737** | 0.730 | 0.830 | 0.800 | **0.560** | **0.642** |

| | **SDXL** | | | | | | | | | | | | |
|---|---|---|---|---|---|---|---|---|---|---|---|---|---|
| | **GenEval[↑]** | | | | | | | **DPGBench[↑]** | | | | | |
| Method | single_obj. | two_obj. | count | colors | position | color_attr. | overall | attr. | entity | global | relation | other | overall |
| CFG | 0.980 | 0.682 | **0.459** | 0.859 | **0.128** | 0.225 | 0.555 | 0.807 | **0.827** | 0.854 | 0.871 | **0.668** | 0.751 |
| CFG++ | 0.972 | 0.691 | **0.453** | 0.872 | **0.128** | 0.225 | 0.556 | 0.797 | 0.821 | **0.872** | **0.875** | 0.644 | 0.744 |
| CCFG | **0.984** | **0.717** | 0.441 | **0.886** | 0.123 | **0.240** | **0.565** | **0.810** | 0.822 | 0.863 | 0.873 | 0.600 | **0.752** |

Table 1: Quantitative results on GenEval and DPGBench. **Bold**: best performance.

contrastive weighting intrinsically attenuates negative guidance when the sample is already far from $c^-$, preventing unnecessary distributional shift.

## 4.2 TEXT-CONDITIONED MODELS

Now we test the performance of CCFG in the context of T2I generation, where the conditioning signal is much complex and intertwined. The main results were obtained from StableDiffusion 1.5(SD1.5) and SDXL (Podell et al., 2024), where additional experiments on a flow-matching model of StableDiffusion 3 (Esser et al., 2024) are in Appendix B.4. See Figure 1 and Figure 11 for the examples of T2I generation scenarios with CCFG using positive and negative prompts.

**Guidance with positive prompts only.** Tab. 1 lists the performance of CFG and CCFG on GenEval and DPGBench, the text-to-image generation benchmarks with given text prompts to follow. CCFG showed similar or better performance to CFG in most of the subtasks, and achieved a slightly higher overall score in both benchmarks and a larger model scale on SDXL.

**Guidance with positive and negative prompts.** Figure 5 illustrates the result of different negative sampling algorithms from SD1.5 with several hand-crafted positive and negative prompt pairs. We observed that due to the characteristic of the overlapping text condition, the calculated guidance scale of DNG was insufficient to faithfully remove the concept in the negative prompt(*e.g.*, cane in case (e)). The exaggerated negative guidance of NP induces sample bias(*e.g.*, negative prompt "strawberries" removed *any* red fruit in case (b)) and even hampers the affinity to the positive prompt(*e.g.*,

| | | StableDiffusion 1.5 | | | | | | | | | |
|---|---|---|---|---|---|---|---|---|---|---|---|
| Method | stand-alone | positive prompt[↑] | | | | negative prompt[↓] | | | | overall[↑] | |
| | | CLIP | IR | HPSv2 | VLM | CLIP | IR | HPSv2 | VLM | $VLM_{all}$ | Aesthetic |
| NP | ✓ | 0.2612 | 0.2967 | 0.2572 | 0.7981 | **0.1257** | **-2.0176** | **0.1637** | **0.0849** | 0.7303 | 5.5309 |
| DNG | ✓ | 0.2702 | 0.5794 | 0.2610 | 0.8454 | 0.1408 | -1.8813 | 0.1753 | 0.1117 | 0.7484 | 5.5947 |
| Perp-Neg | | **0.2724** | **0.6219** | 0.2642 | 0.8462 | 0.1364 | -1.9037 | 0.1798 | 0.0943 | 0.7664 | 5.6017 |
| SLD | | 0.2687 | 0.5640 | 0.2622 | 0.8485 | 0.1346 | -1.9184 | 0.1715 | 0.0931 | 0.7695 | 5.6053 |
| SAFREE | | 0.2658 | 0.4329 | 0.2595 | 0.8272 | 0.1341 | -1.9551 | 0.1709 | 0.0909 | 0.7470 | 5.5969 |
| CCFG | ✓ | 0.2703 | 0.5791 | **0.2695** | **0.8540** | 0.1339 | -1.9356 | 0.1709 | 0.0894 | **0.7777** | **5.6211** |
| | | SDXL | | | | | | | | | |
| NP | ✓ | 0.2665 | 0.9658 | 0.2851 | 0.8902 | **0.1313** | -1.9254 | 0.1704 | 0.1093 | 0.7929 | 5.9190 |
| DNG | ✓ | **0.2745** | 1.1652 | 0.2936 | 0.9040 | 0.1456 | -1.7628 | 0.1842 | 0.1386 | 0.7787 | 5.9891 |
| Perp-Neg | | 0.2735 | **1.1919** | 0.2976 | 0.9158 | 0.1384 | -1.8324 | 0.1792 | 0.1192 | 0.8066 | 6.0155 |
| SLD | | 0.2733 | 1.1743 | 0.2957 | **0.9185** | 0.1392 | -1.8179 | 0.1802 | 0.1150 | **0.8128** | 6.0125 |
| SAFREE | | 0.2729 | 1.1831 | 0.2962 | 0.9160 | 0.1389 | -1.8529 | 0.1781 | 0.1182 | 0.8077 | 5.9987 |
| CCFG | ✓ | 0.2739 | 1.1840 | **0.2978** | 0.9164 | 0.1381 | -1.8234 | 0.1790 | 0.1139 | 0.8120 | **6.0158** |

Table 2: Quantitative results on PIE-Bench-Neg. **Bold**: best performance, underline: second best.

"tree" and "traveler" in cases (d) and (e)). We also observed that the quality of the images can be heavily degraded, with unrecognizable contents and artifacts for cases (a) and (c). While Perp-Neg and SLD can integrate negative and positive guidance to reduce conflicts with the positive prompt, they may still fail to suppress unwanted features without careful hyperparameter search. CCFG has successfully eliminated the concept of negative prompts while maintaining positive prompts and image quality. See Appendix B.4 for a quantitative analysis of these case studies.

For a more thorough evaluation beyond case studies, we tested the behavior of CCFG and other baselines on carefully designed benchmarks. Nonetheless, very few benchmarks provide or evaluate the output image with both positive and negative prompts; most focus on how positive prompts are suppressed by the opposing negative prompts (Schramowski et al., 2023).Therefore, we designed to build a negative prompt that "*overlaps but doesn't conflict*" with its corresponding positive prompt, based on source and target prompt pairs from known image editing benchmarks. Specifically, for given source and target prompt pairs in PIE-Bench (Ju et al., 2024), we compare the two prompts and utilize their mismatching objects, features, or style. This disagreement is utilized as a negative prompt, and its corresponding positive prompt is set to one of the two original prompts that doesn't contain the disagreement. We refer to this dataset as *PIE-Bench-Neg*, where a more detailed construction process can be found in Appendix D.3.

We used various text-alignment metrics of CLIP cosine similarity (Radford et al., 2021), ImageReward (Xu et al., 2024), and HPS-v2 (Wu et al., 2023) to evaluate the faithfulness to the given positive or negative prompts. However, these metrics have the limitation of conflating image quality, prompt alignment, and human preference in a single value(*e.g.*, the score with negative prompts can decrease by not the removal of unwanted features, but simply image corruption). To address this limitation, we adopt the VLM-as-a-judge framework (Lee et al., 2024; Chen et al., 2024) to perceptually evaluate the prompt alignment on a scale of 0 to 1 (See Appendix D.3 for details). To solely quantify the quality of the image, we measured the Aesthetic Score of the samples.

Tab. 2 shows the performance of CCFG and other baselines on our PIE-Bench-Neg benchmark. DNG shows the highest alignment scores with the negative prompt due to weak negative guidance, and NP shows degradation in both alignment with the positive prompt and the Aesthetic score. The drawback of these two methods led to lower $VLM_{all}$, which is the accuracy for both positive prompt alignment and negative prompt removal according to the VLM. Meanwhile, CCFG had a better trade-off between two different guidance, and achieves more output images that perceptually satisfy the given conditions with a better image quality; overall, it showed comparable or better performance with empirical baselines of Perp-Neg and SLD. We conclude that CCFG stands as a flexible and competitive method in the T2I generation task, on top of its broader applicability for negative-only guidance as discussed in Section 4.1.

# 5 CONCLUSION

In this work, we proposed CCFG, a sampling method to guide diffusion samples to satisfy or repel the given condition by optimizing the denoising direction with contrastive loss. Through the experiments on various datasets, we showed that CCFG resembles the positive guidance of the tra-

ditional CFG and resolves the downsides of the previous negative guidance methods, resulting in high-quality samples while faithfully aligning or avoiding specific attributes. Despite its effective computational overload and performance, one limitation could be the absence of an analytic closed-form sampling distribution corresponding to the proposed CCFG sampling. It would be a possible future work to propose negative guidance that allows a probabilistic interpretation while preserving the suggested advantages of CCFG. We believe that CCFG can be easily integrated and benefit various downstream applications and modalities that require negative sampling.

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

---

**Algorithm 2** Deterministic sampling on flow-matching models with CCFG

---

**Require:** positive prompt $c^+$, negative prompt $c^-$, $\{\omega_t\}_{t=0}^1 > 0$, $\{\tau_t\}_{t=0}^1 > 0$
1: Initialize $\boldsymbol{x}_t \sim \mathcal{N}(0, \mathbf{I})$
2: **for** $i = 0$ **to** 1 **do**
3:      $\lambda^+ = \frac{2}{1 + e^{-\tau_t \|\hat{\boldsymbol{v}}_\varnothing(\boldsymbol{x}_t) - \hat{\boldsymbol{v}}_{c^+}(\boldsymbol{x}_t)\|^2}}$ **if** $c^+$ != *None* **else** 0
4:      $\lambda^- = \frac{2e^{-\tau_t \|\hat{\boldsymbol{v}}_\varnothing(\boldsymbol{x}_t) - \hat{\boldsymbol{v}}_{c^-}(\boldsymbol{x}_t)\|^2}}{1 + e^{-\tau_t \|\hat{\boldsymbol{v}}_\varnothing(\boldsymbol{x}_t) - \hat{\boldsymbol{v}}_{c^-}(\boldsymbol{x}_t)\|^2}}$ **if** $c^-$ != *None* **else** 0
5:      $\hat{\boldsymbol{v}}_c^{\omega_t}(\boldsymbol{x}_t) = \hat{\boldsymbol{v}}_\varnothing(\boldsymbol{x}_t) + \omega_t \lambda^+ (\hat{\boldsymbol{v}}_{c^+}(\boldsymbol{x}_t) - \hat{\boldsymbol{v}}_\varnothing(\boldsymbol{x}_t)) - \omega_t \lambda^- (\hat{\boldsymbol{v}}_{c^-}(\boldsymbol{x}_t) - \hat{\boldsymbol{v}}_\varnothing(\boldsymbol{x}_t))$
6:      $\boldsymbol{x}_t = \boldsymbol{x}_t + \hat{\boldsymbol{v}}_c^{\omega_t}(\boldsymbol{x}_t)dt$
7: **end for**
8: **return** $\boldsymbol{x}_t$

---

**Algorithm 3** DDIM deterministic sampling with CCFG guidance on the posterior mean

---

**Require:** $\{\rho_t\}_{t=1}^T > 0$, $\{\tau_t\}_{t=1}^T > 0$, positive prompt $c^+$, negative prompt $c^-$
1: Initialize $\boldsymbol{x}_t \sim \mathcal{N}(0, \mathbf{I})$
2: **for** $i = T$ **to** 1 **do**
3:      $\lambda^+ = \frac{2}{1 + e^{-\tau_t \|\hat{\boldsymbol{\epsilon}}_\varnothing(\boldsymbol{x}_t) - \hat{\boldsymbol{\epsilon}}_{c^+}(\boldsymbol{x}_t)\|^2}}$ **if** $c^+$ != *None* **else** 0
4:      $\lambda^- = \frac{2e^{-\tau_t \|\hat{\boldsymbol{\epsilon}}_\varnothing(\boldsymbol{x}_t) - \hat{\boldsymbol{\epsilon}}_{c^-}(\boldsymbol{x}_t)\|^2}}{1 + e^{-\tau_t \|\hat{\boldsymbol{\epsilon}}_\varnothing(\boldsymbol{x}_t) - \hat{\boldsymbol{\epsilon}}_{c^-}(\boldsymbol{x}_t)\|^2}}$ **if** $c^-$ != *None* **else** 0
5:      $\hat{\boldsymbol{\epsilon}}_c^{\rho_t}(\boldsymbol{x}_t) = \hat{\boldsymbol{\epsilon}}_\varnothing(\boldsymbol{x}_t) + \rho_t \lambda^+ (\hat{\boldsymbol{\epsilon}}_{c^+}(\boldsymbol{x}_t) - \hat{\boldsymbol{\epsilon}}_\varnothing(\boldsymbol{x}_t)) - \rho_t \lambda^- (\hat{\boldsymbol{\epsilon}}_{c^-}(\boldsymbol{x}_t) - \hat{\boldsymbol{\epsilon}}_\varnothing(\boldsymbol{x}_t))$
6:      $\hat{\boldsymbol{x}}_c^{\rho_t}(\boldsymbol{x}_t) = (\boldsymbol{x}_t - \sqrt{1 - \bar{\alpha}_t}\hat{\boldsymbol{\epsilon}}_c^{\rho_t}(\boldsymbol{x}_t))/\sqrt{\bar{\alpha}_t}$
7:      $\boldsymbol{x}_{t-1} = \sqrt{\bar{\alpha}_{t-1}}\hat{\boldsymbol{x}}_c^{\rho_t}(\boldsymbol{x}_t) + \sqrt{1 - \bar{\alpha}_{t-1}}\hat{\boldsymbol{\epsilon}}_\varnothing(\boldsymbol{x}_t)$
8: **end for**
9: **return** $\boldsymbol{x}_t$

---

## A    CCFG ON FLOW-MATCHING MODELS

Our derivation of CCFG leverages the distribution of the next denoised sample with different conditions, which came from the stochastic sampling process, On the other hand, recent text-to-image models such as StableDiffusion 3 (Esser et al., 2024) are constructed within the flow-matching framework, where generation is modeled as the solution of an Ordinary Differential Equation(ODE) driven by a learned velocity field. Unlike diffusion models based on stochastic forward–reverse processes, flow-matching models employ deterministic dynamics of (22) over the time interval $[0, 1]$ that transport Gaussian noise toward the data distribution.

$$d\boldsymbol{x}_t = \boldsymbol{v}_\theta(\boldsymbol{x}_t, t)dt, \quad \boldsymbol{x}_0 \sim \mathcal{N}(0, \boldsymbol{I}). \tag{22}$$

That said, according to the Fokker-Planck equation, there exists a set of stochastic differential equations that has the same marginal distribution $p(\boldsymbol{x}_t)$ as (22),

$$d\boldsymbol{x}_t = \left( \boldsymbol{v}_\theta(\boldsymbol{x}_t, t) + \frac{t\sigma_t^2}{2(1-t)} \left( \boldsymbol{v}_\theta(\boldsymbol{x}_t, t) - \frac{1}{t}\boldsymbol{x}_t \right) \right) dt + \sigma_t d\mathbf{B}_t, \quad \boldsymbol{x}_0 \sim \mathcal{N}(0, \mathbf{I}), \tag{23}$$

where $\sigma_t$ is an arbitrary choice of time-dependent diffusion coefficient. The stochasticity in (23) enables the same analogy that we derived CCFG in diffusion models in Section 3.2. Algorithm 2 describes the deterministic sampling process on flow-matching models with positive and negative prompt guidances by CCFG.

We've conducted the same experiments as those done in Section 4.2, where we tested CCFG on GenEval, DPGBench, and PIE-Bench-Neg. Note that since other baselines in the main manuscript were not proposed for the flow-matching model, we compared CCFG with a naive CFG, NP, and CFG-zero (Fan et al., 2025) as an alternative for CFG on flow-matching models. Table 3 and Table 4 show that CCFG achieved both positive prompt alignment and negative prompt removal with a

similar trend in diffusion models, demonstrating the generalizability of CCFG across different model types. See Figure 11 for the example of CCFG sampling on StableDiffusion 3.

# B  ADDITIONAL RESULTS

## B.1  CCFG IN PERSPECTIVE OF GUIDED SAMPLING ON THE POSTERIOR MEAN

In the suggested algorithm of CCFG in Alg. 1, we applied the suggested contrastive loss to optimize the denoising direction $\hat{\boldsymbol{\epsilon}}_\varnothing(\boldsymbol{x}_t)$, with the same spirit of the conventional CFG that modifies $\hat{\boldsymbol{\epsilon}}_\varnothing(\boldsymbol{x}_t)$ directly. This enabled us to conduct a fair comparison with the conventional CFG using an equal guidance scale.

Meanwhile, recall that the motivation for constructing CCFG was to pose positive and negative prompt sampling as guided sampling similar to CFG++ (Chung et al., 2024b), which optimizes the posterior mean $\hat{\boldsymbol{x}}_\varnothing(\boldsymbol{x}_t)$ with the iteration of (7). In order to follow this scheme, one would have to use the null noise $\hat{\boldsymbol{\epsilon}}_\varnothing(\boldsymbol{x}_t)$ in the renoising process, as shown in lines 9∼10 of Alg. 3.

Here, we show that these two different implementations can be equivalent with appropriate guidance scale scheduling. Considering a single DDIM reverse sampling step, the obtained $\boldsymbol{x}_{t-1}$ from $\boldsymbol{x}_t$ by Alg. 3 can be written as follows:

$$\boldsymbol{x}_{t-1} = \frac{1}{\sqrt{\alpha_t}}\boldsymbol{x}_t + \left(\sqrt{1-\bar{\alpha}_{t-1}} - \frac{\sqrt{1-\bar{\alpha}_t}}{\sqrt{\alpha_t}}\right)\hat{\boldsymbol{\epsilon}}_\varnothing - \rho_t\lambda\frac{\sqrt{1-\bar{\alpha}_t}}{\sqrt{\alpha_t}}(\hat{\boldsymbol{\epsilon}}_c - \hat{\boldsymbol{\epsilon}}_\varnothing). \tag{24}$$

In contrast, in Alg. 1, the iteration reads

$$\boldsymbol{x}_{t-1} = \frac{1}{\sqrt{\alpha_t}}\boldsymbol{x}_t + \left(\sqrt{1-\bar{\alpha}_{t-1}} - \frac{\sqrt{1-\bar{\alpha}_t}}{\sqrt{\alpha_t}}\right)\hat{\boldsymbol{\epsilon}}_\varnothing - \omega_t\lambda\left(\sqrt{1-\bar{\alpha}_{t-1}} - \frac{\sqrt{1-\bar{\alpha}_t}}{\sqrt{\alpha_t}}\right)(\hat{\boldsymbol{\epsilon}}_c - \hat{\boldsymbol{\epsilon}}_\varnothing). \tag{25}$$

Therefore, by setting

$$\rho_t = \omega_t\left(1 - \frac{\sqrt{\alpha_t}\sqrt{1-\bar{\alpha}_{t-1}}}{\sqrt{1-\bar{\alpha}_t}}\right), \tag{26}$$

we can show the equivalence between Alg. 1 and Alg. 3. Hence, for a direct and fair comparison of CCFG against conventional CFG, we chose CCFG with Alg. 1 as our implementation.

## B.2  DERIVATION OF THE GUIDANCE TERM FROM CCFG OBJECTIVE FUNCTION

Note that

$$f(\boldsymbol{x}; \boldsymbol{a}, \boldsymbol{b}, \tau_t) := -\log\frac{e^{-\tau_t\|\boldsymbol{x}-\boldsymbol{a}\|_2^2}}{e^{-\tau_t\|\boldsymbol{x}-\boldsymbol{a}\|_2^2} + e^{-\tau_t\|\boldsymbol{x}-\boldsymbol{b}\|_2^2}} \tag{27}$$

$$\Leftrightarrow \nabla_{\boldsymbol{x}}f = -\frac{2\tau_t(\boldsymbol{a}-\boldsymbol{b})e^{-\tau_t\|\boldsymbol{x}-\boldsymbol{b}\|_2^2}}{e^{-\tau_t\|\boldsymbol{x}-\boldsymbol{a}\|_2^2} + e^{-\tau_t\|\boldsymbol{x}-\boldsymbol{b}\|_2^2}} \tag{28}$$

When we put $(\boldsymbol{x} = \hat{\boldsymbol{\epsilon}}_\varnothing, \boldsymbol{a} = \hat{\boldsymbol{\epsilon}}_c, \boldsymbol{b} = \hat{\boldsymbol{\epsilon}}_\varnothing)$ and $(\boldsymbol{x} = \hat{\boldsymbol{\epsilon}}_\varnothing, \boldsymbol{a} = \hat{\boldsymbol{\epsilon}}_\varnothing, \boldsymbol{b} = \hat{\boldsymbol{\epsilon}}_c)$ into (28), we get the positive and negative CCFG guidance term for (18) and (21), respectively.

## B.3  ANALYSIS ON THE SELECTION OF CCFG HYPERPARAMETERS

In the process of simplifying (15) into (16), the explicit expression of $\tau_t$ in terms of $\sigma_t$ can be obtained as

$$\frac{\|\boldsymbol{\mu}(\boldsymbol{x}_t, \boldsymbol{\epsilon}) - \boldsymbol{\mu}(\boldsymbol{x}_t, \hat{\boldsymbol{\epsilon}}_c)\|_2^2}{2\sigma_t^2} = \frac{\left(\frac{\sqrt{\bar{\alpha}_{t-1}}\sqrt{1-\bar{\alpha}_t}}{\sqrt{\bar{\alpha}_t}} - \sqrt{1-\bar{\alpha}_{t-1}-\sigma_t^2}\right)^2}{2\sigma_t^2}\|\boldsymbol{\epsilon} - \hat{\boldsymbol{\epsilon}}_c\|_2^2 \tag{29}$$

$$:= \tau_t\|\boldsymbol{\epsilon} - \hat{\boldsymbol{\epsilon}}_c\|_2^2 \tag{30}$$

$$\Leftrightarrow \tau_t := \frac{\left(\frac{\sqrt{\bar{\alpha}_{t-1}}\sqrt{1-\bar{\alpha}_t}}{\sqrt{\bar{\alpha}_t}} - \sqrt{1-\bar{\alpha}_{t-1}-\sigma_t^2}\right)^2}{2\sigma_t^2} \tag{31}$$

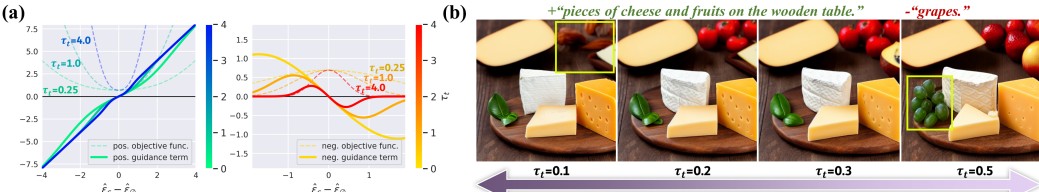

Figure 6: (a) The objective function and their guidance term of CCFG with different selection of $\tau_t$, in positive guidance (left) and negative guidance (right). The guidance scale $\hat{\epsilon}_c^{\omega_t}$ is fixed to 1.0. (b) The examples of CCFG sampling with different $\tau_t$, using SD1.5 and the same initial noise.

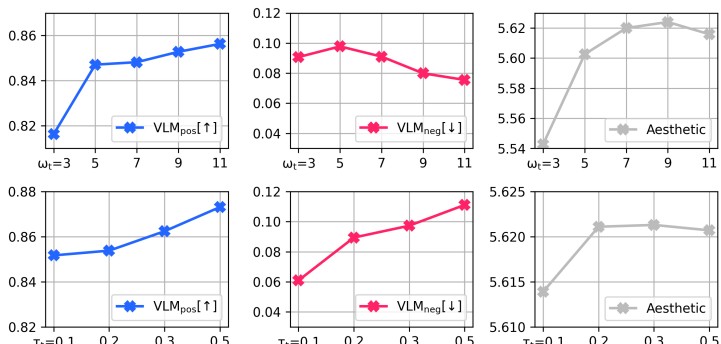

Figure 7: Performance of CCFG on PIE-Bench-Neg with different selection of $\omega_t$(top row) and $\tau_t$(bottom row).

where $0 \leq \sigma_t^2 \leq 1 - \bar{\alpha}_{t-1}$.

Figure 6-(a) shows the objective functions and the corresponding guidance terms of CCFG as $\tau_t$ changes. In any $\tau_t$, the guidance term for positive CCFG approximates a linear function for the original CFG, when $\|\hat{\epsilon}_c - \hat{\epsilon}_\varnothing\|$ is sufficiently large or near 0. The guidance coefficient $\lambda$ for the positive prompt becomes constant when $\tau_t$ is 0 or diverges to infinity, making positive CCFG equivalent to the original CFG. In the case of the negative guidance, $\lambda$ becomes 0 when $\tau_t$ diverges to infinity, and no negative guidance is applied. As $\tau_t$ goes to 0, $\lambda$ again becomes constant, making negative CCFG equivalent to NP. In general, the choice of $\tau_t$ affects more on the negative CCFG since it governs the range of $\|\hat{\epsilon}_c - \hat{\epsilon}_\varnothing\|$ in which the negative guidance is applied sufficiently.

Figure 6-(b) illustrates this behavior in T2I generation with SD1.5, where too large $\tau_t$ disables proper negative guidance and too small $\tau_t$ introduces the drawback of NP(*e.g.* applying too much negative guidance disrupts following the positive prompt).

Figure 7 shows the performance of CCFG on PIE-Bench-Neg with different selection of $\omega_t$ and $\tau_t$. The positive prompt alignment(VLM$_{pos}$) and negative prompt removal(VLM$_{neg}$) improve as the guidance scale $\omega_t$ increases, while the overall quality of the image(Aesthetic Score) has a sweet spot at a suitable $\omega_t$. This trend agrees with a well-known observation of traditional CFG. Meanwhile, as illustrated in Figure 6-(b), a smaller $\tau_t$ can more thoroughly remove the negative concept, but also hinders the positive prompt alignment.

### B.4 TEXT-TO-IMAGE SAMPLING WITH POSITIVE AND NEGATIVE PROMPTS

Figure 8 shows the quantitative results on five T2I case studies in Figure 5 with CLIP cosine similarity score, ImageReward, HPS-v2, and perceptual accuracy judged by VLM. Here, we show the average score over the five cases for each method. We sampled 200 images for each method and case. The ideal behavior as an effective and safe negative sampling method is to maintain alignment with positive prompts while decreasing alignment with negative prompts, compared to when no negative guidance is applied.

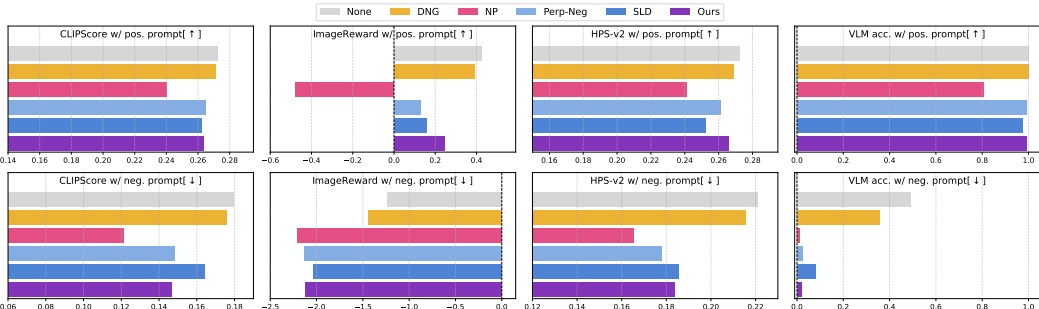

Figure 8: Quantitative evaluation results over the case scenarios in Figure 5. "None" denotes cases where only the positive guidance is applied.

| | StableDiffusion 3 | | | | | | | | | | | | |
|---|---|---|---|---|---|---|---|---|---|---|---|---|---|
| | GenEval[↑] | | | | | | | DPGBench[↑] | | | | | |
| Method | single_obj. | two_obj. | count | colors | position | color_attr. | overall | attr. | entity | global | relation | other | overall |
| CFG | 0.984 | 0.740 | 0.559 | 0.823 | 0.237 | 0.449 | 0.632 | 0.861 | 0.878 | **0.844** | **0.910** | 0.780 | 0.825 |
| CCFG | **0.988** | 0.748 | **0.575** | **0.840** | 0.250 | 0.463 | **0.644** | **0.865** | **0.883** | 0.839 | 0.906 | **0.816** | **0.827** |

Table 3: Quantitative results with StableDiffusion 3 on GenEval and DPGBench. **bold**: best performance.

DNG showed the smallest alignment metric decrease between its samples and the given positive prompt, but failed to thoroughly rule out the features in the negative prompt; the alignment with the negative prompt was marginally decreased, and VLM often detected the attributes described in the negative prompt. Meanwhile, NP showed the lowest alignment score not only with the negative prompts, but also for the positive prompts. Also, almost 20% of its output was judged to not satisfy the positive prompt by VLM. While Perp-Neg and SLD can orchestrate the negative guidance with the positive guidance to prevent the misalignment between the positive prompt, they also occasionally fail to remove the unwanted features without careful hyperparameter search. Our proposed CCFG removes the undesired properties enough to be undetected by VLM, and still maintains the agreement with the positive prompts.

In Figure 11, we display additional image generation examples by CCFG and given positive-negative prompt pairs from StableDiffusion 1.5, SDXL, and StableDiffusion 3. Leveraging the ability of CCFG for an effective and stable concept negation, we can resolve certain undesirable features such as objects, motions, and multiple composed features. Furthermore, CCFG can help the model generate minority samples and remove potentially harmful objects by negating potential internal bias or inappropriate contents.

### B.5 Possible failure cases

Figure 9 presents several failure cases of CCFG on text-to-image generation tasks; when the negative condition involves multiple attributes simultaneously, CCFG might suppresses only a subset of the undesired aspects. And when the undesired negative prompt describes subtle continuous attributes, the negative guidance strength of CCFG may be weakened in the process of concept removal, resulting in outputs on the borderline of unwanted features.

## C The behavior of NP on different diffusion sampling scenarios

So far, we've shown the theoretical pitfall of the NP and demonstrated its negative effect on the sample distribution and its quality. Meanwhile, we observed that the criticality of NP's drawback has been mitigated as the learned data distribution and the nature of the used condition get more complex: the NP term completely shifts the sample distribution out from the marginal support in the toy example of Figure 2, but often produced reasonable images for T2I generation tasks. Indeed, many off-the-shelf implementations for the T2I generation model have used this guidance to this day for concept negation (Gandikota et al., 2023; Koulischer et al., 2025; Armandpour et al., 2023; Liu et al., 2022).

| StableDiffusion 3 | | | | | | | | | |
|---|---|---|---|---|---|---|---|---|---|
| | positive prompt[↑] | | | | negative prompt[↓] | | | | overall[↑] | |
| Method | CLIP | IR | HPSv2 | VLM | CLIP | IR | HPSv2 | VLM | $VLM_{all}$ | Aesthetic |
| NP | 0.2616 | 1.0302 | 0.2813 | 0.9415 | 0.1335 | **-1.9418** | **0.1722** | **0.1135** | 0.8346 | 5.7214 |
| CCFG | **0.2722** | **1.2976** | **0.2891** | **0.9630** | **0.1316** | -1.7148 | 0.1958 | 0.1185 | **0.8489** | **5.7751** |

Table 4: Quantitative results with StableDiffusion 3 on PIE-Bench-Neg. **bold**: best performance.

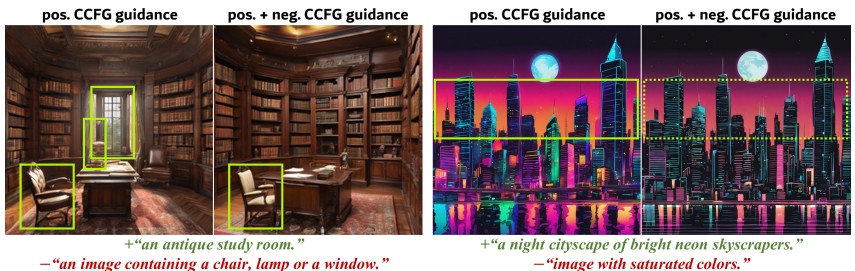

Figure 9: Examples of failure cases on text-to-image generation with positive and negative prompts via CCFG.

We insist that this phenomenon occurs for the following reasons: First, as the possible conditions become more diverse and their relationship with the data distribution becomes more complex, it becomes harder for the model to accurately parameterize the entire unconditional and conditional distribution. It is well known that many other T2I models perform relatively poorly for pure unconditional generation, and extra distribution sharpening with CFG is almost essential for acceptable image quality. When this happens, even an unbounded guidance term can enhance the sample, until it overshoots and introduces artifacts. For more simplified scenarios like MNIST, where the model can accurately learn the target data distribution, the downside of NP becomes more evident. The second reason comes from $p(\boldsymbol{c}^-|\boldsymbol{x})^{-\gamma}$, the factor that NP introduced to the sampling distribution. In a class-conditioned dataset where each class doesn't overlap too much, $p(\boldsymbol{c}^-|\boldsymbol{x})$ can be close to 1 if the given image is sampled from class $\boldsymbol{c}^-$. This contrasts $p(\boldsymbol{c}^-|\boldsymbol{x})^{-\gamma}$ with different $\boldsymbol{x}$ significantly and assigns a high probability for the region with low $p(\boldsymbol{c}^-|\boldsymbol{x})$, even for $\boldsymbol{x}$ with low marginal probability. On the other hand, a single image can be described by numerous different texts, therefore the magnitude of $p(c|\boldsymbol{x})$ for an image-text pair is small no matter how well $c$ describes the given image.

Nonetheless, we believe that the fundamental flaws of NP have been already demonstrated with various experiments, and a risk of failure still remains for T2I models.

# D EXPERIMENTAL DETAILS

## D.1 SAMPLING HYPERPARAMETERS

Tab. 5 lists the hyperparameter values used for DNG and CCFG for the results in the main manuscript. Apart from the guidance scale $\omega$ that is shared by all negative sampling methods we've tested, DNG requires a prior for the given condition $p(c)$, an affine transformation weight $\tau'$ and bias $\delta$ for its guidance term calculation. We first took the values from the authors of DNG (Koulischer et al., 2025) as a basis, then additionally searched and tuned for the best performance on each dataset. We note that DNG only showed acceptable performances with the heavily exaggerated condition prior $p(c)$ which are far from the reasonable value(*e.g.* 0.1 for MNIST and CIFAR10).

## D.2 CLASS-CONDITIONED MODELS

For the experiments about CCFG on MNIST and CIFAR10, we trained DDPM (Ho et al., 2020) for each dataset from scratch which enables both conditional and unconditional generation to perform CFG, CCFG, and other sampling methods. The external classifier was imported from the following checkpoints, where the ViT-base is trained with the classification of MNIST and CIFAR10, respec-

| Dataset | DNG | SLD | CCFG |
|---|---|---|---|
| MNIST | $p(c)$=0.25, $\tau'$=0.25, $\delta$=0 | (N/A) | $\tau$=1.0 |
| CIFAR10 | $p(c)$=0.5, $\tau'$=0.1, $\delta$=0.0002 | (N/A) | $\tau$=0.25 |
| ImageNet-1k | $p(c)$=0.01, $\tau'$=0.2, $\delta$=0.0002 | $\delta$=15, $s_S$=200, $\lambda$=0.00, $s_m$=0.00, $\beta_m$=0.4 | $\tau$=0.5 |
| StableDiffusion 1.5 | $p(c)$=0.01, $\tau'$=0.2, $\delta$=0.003 | $\delta$=10, $s_S$=1000, $\lambda$=0.01, $s_m$=0.3, $\beta_m$=0.4 | $\tau$=0.2 |
| SDXL | $p(c)$=0.01, $\tau'$=0.2, $\delta$=0.003 | $\delta$=10, $s_S$=1000, $\lambda$=0.01, $s_m$=0.3, $\beta_m$=0.4 | $\tau$=0.1 |
| StableDiffusion 3 | (N/A) | (N/A) | $\tau$=0.005 |

Table 5: Sampling hyperparameters used for DNG, SLD, and CCFG.

tively. The diffusion models were trained with 500 noise timesteps. For the inference, we used deterministic DDIM sampling with NFE=50.

For the experiment on ImageNet-1k, We prepared 100 class pairs with strong visual proximity $(c^+, c^-)$ (e.g., *Maltese_dog - Shih-Tzu, Italian_greyhound - whippet, Samoyed - Pomeranian, racer - sports_car, beach_wagon - cab, bathing_cap - shower_cap, garden_spider - wolf_spider*) according to ImageNet hierarchy Bostock (2019) and the following pipeline:

- For each ImageNet-1k class, we retrieve the corresponding WordNet ID (wnid) and its hypernyms.
- We construct a pool of candidate pairs by including all pairs of classes that share an identical hypernym.
- From this pool, we uniformly sample 100 class pairs to obtain the final benchmark set, while ensuring that every class in the 100 class pairs is unique.

The sampling was done at $256\times256$ using an ImageNet-1k–pretrained DiT-XL/2 (Peebles & Xie, 2023) by deterministic DDIM sampling with NFE=50. For the external classifier, we used a frozen ResNet50 (He et al., 2015) pretrained on ImageNet-1k using torchvision's default IMA-GENET1K_V2 weights.

## D.3 TEXT-CONDITIONED MODELS

For the SD1.5 and SDXL image sampling, we used DDIM deterministic sampling with 50 NFE and a guidance scale of 7.5. In the flow-matching model sampling with StableDiffusion 3, we used a deterministic sampling with 28 NFE and a guidance scale of 3.0. In the case of DNG and SLD, we used one of the hyperparameter configurations provided by the authors.

To build the PIE-Bench-Neg benchmark, we processed each pair of source and target prompts in PIE-Bench (Ju et al., 2024) according to the given affiliation that describes the editing task:

- add_object: Compared to the source prompt(*e.g., "a bee illustration on a paper"*), the newly added objects in the target prompt are listed in the attribute of aspect_mapping(*e.g., "leaves, flower"*). These objects become the negative prompt, and the source prompt is the positive prompt.
- delete_object: The objects from the source prompt(*e.g., "a wathet bird sitting on a branch of yellow flowers"*) removed in the target prompt(*e.g., "a wathet bird sitting on a branch"*) are listed in the attribute of aspect_mapping(*e.g., "flowers"*). These objects become the negative prompt, and the target prompt is the positive prompt.
- change_object, change_attribute_content, change_background: The changed appearances from the source prompt(*e.g., "fishes and kelp in the ocean"*) are marked in the target prompt(*e.g., "[sharks] and [flowers] in the ocean"*), which becomes the negative prompt(*e.g., "sharks, flowers"*). The source prompt is used for the positive prompt.
- change_attribute_pose, change_attribute_color, change_attribute_material: These editing tasks require certain features of the objects in the source prompt(*e.g., "a brown bear is sleeping with a sign that says no sleep"*) to be changed, and the mapping between them can be found in key-value pairs of aspect_mapping(*e.g., "bear":"black", "sign":"white"*). We combine the objects and

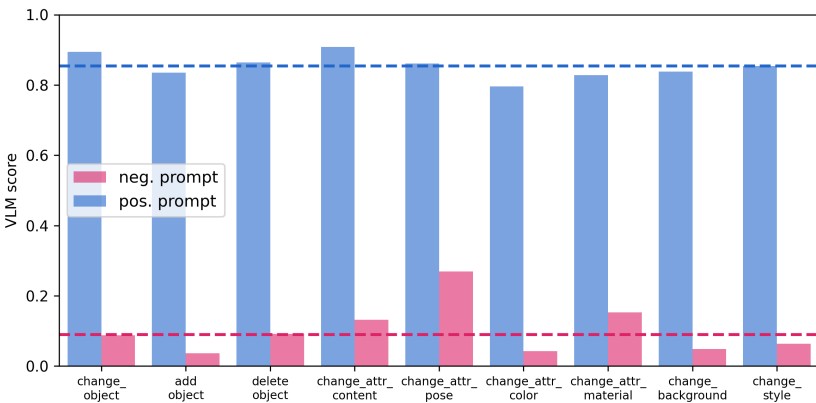

Figure 10: The performance of CCFG in PIE-Bench-Neg prompts from different PIE-Bench categories. The dotted blue and pink line represents the overall mean VLM score for positive and negative prompts, respectively.

their target editing features to construct a negative prompt(*e.g., "black bear, white sign"*), and the original source prompt is used as the positive prompt.

- change_style: The target style is described in the attribute of blended_words(*e.g., "baroque, watercolor"*). These styles become the negative prompt, and the source prompt(*e.g., "a cartoon giraffe sitting down on a white background"*) is the positive prompt.

In total, we prepared 560 positive prompts, paired with the corresponding negative prompt that is correlated but not entirely opposed to the positive prompt.

The performance of CCFG in PIE-Bench-Neg prompts from each of these category is shown in Figure 10. Here, we observed that CCFG is relatively less effective on the series of *"change_attribute_XX"* categories, which tend to yield relatively lower positive prompt VLM scores(*e.g.*, *change_attr_color*) or higher negative prompt VLM scores(*e.g., change_attr_pose*). This can be because the negative prompts from these categories require removing specific attributes of something explicitly contained in the positive prompt, therefore conveying overlapping concepts and making the task more difficult. We also found that having nouns as negative prompts(*e.g., change_object*) or adjectives(*e.g., change_style*) shows similar performances.

To evaluate a perceptual agreement of the image with the given prompt, separated from the image quality, we employed an open-source VLM Gemma3-27B (Team, 2025). We utilize Gemma3 to decompose the given prompt and check the given image's agreement with each component to eventually rate the image. Prompt 1 was used to evaluate the perceptual accuracy of the given image to follow the text prompt using VLM. The last line of the output is interpreted as a fraction between 0 and 1, where 1 is the highest possible agreement score. Similar to how we calculated the overall accuracy in Section 4.1, we multiply the VLM score for the positive and the negative prompt as $VLM_{all}$ to quantify how the model output matches for both prompt conditions.

# E   THE USE OF LARGE LANGUAGE MODELS (LLMS)

LLMs were not involved in research ideation, methodological design, or the writing of the manuscript. The authors retain full responsibility for all scientific content.

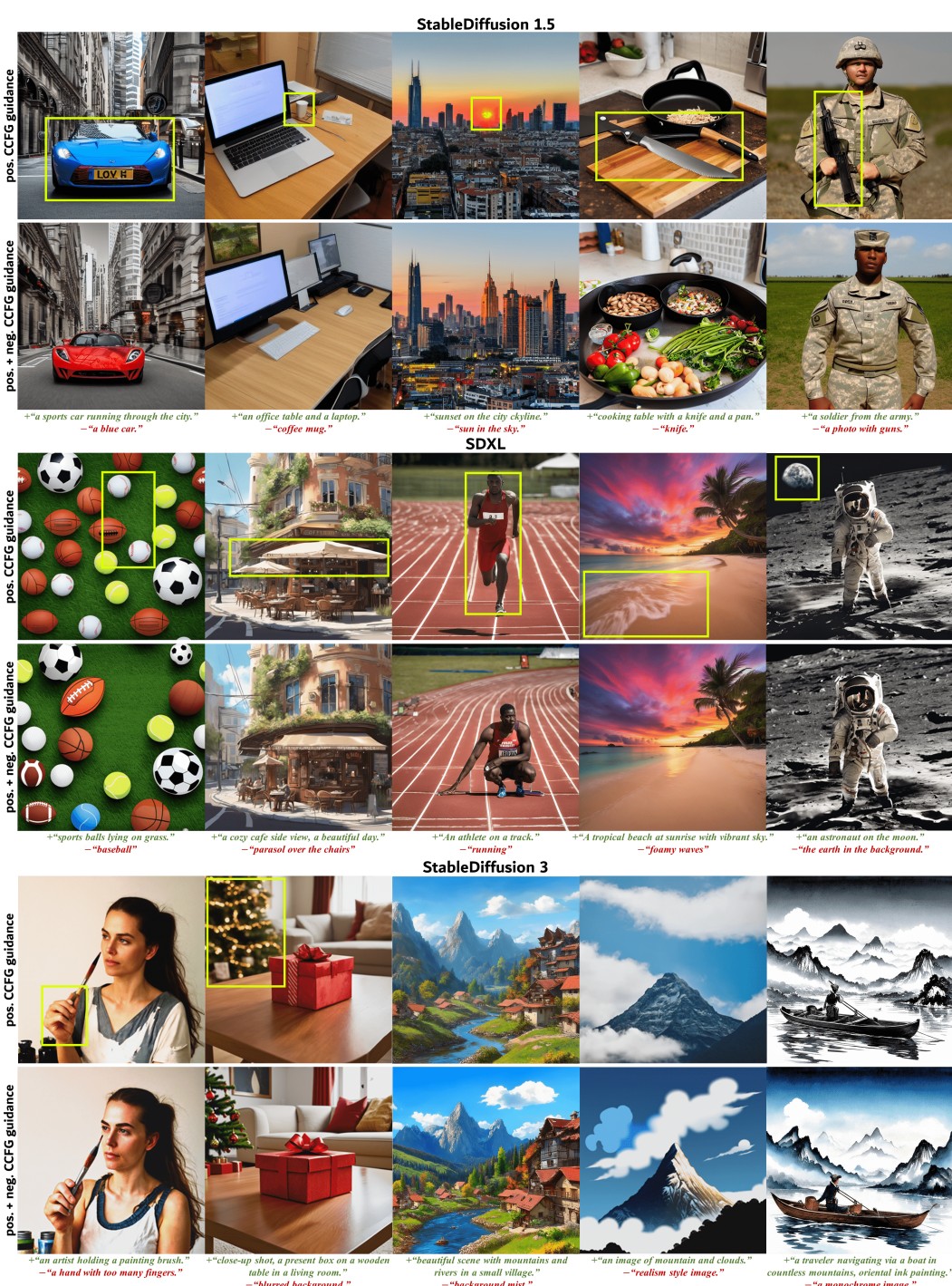

Figure 11: Examples of text-to-image generation results with positive and negative prompts via CCFG, sampled from StableDiffusion 1.5, SDXL, and StableDiffusion 3. In each model, the samples in the same column were sampled from the same initial noise.

**prompt 1**

I'll give you an image and a text prompt. You have to evaluate how well the image satisfies the prompt. In the process, you first list all the objects and appearances(with its corresponding features) described in the prompt. Don't count the word "image" in the prompt as an object. Then, you observe the image and decide whether each detected object/appearance in the prompt is contained in the image.

The last line of your response should be a format of "Final answer: a over b". Here, "a" is the number of objects that are both contained in the prompt and the image, and "b" is the total number of objects in the prompt. Here is the example:

If the prompt is "an image with a blue cat and a red dog" and the image contains a blue cat and a yellow dog, then the prompt contains "a blue cat" and "a red dog". "A blue cat" is contained in the image, and "a red dog" is not contained in the image. Therefore, the answer should be "1 over 2".

Now, the input sentence is "<caption>". Observed the given image and answer.

