# OpenReview forum: "Contrastive CFG: Guiding Diffusion Sampling by Contrasting Positive and Negative Concepts"
_ICLR.cc/2026/Conference — Submitted to ICLR 2026_

### Official Review · Reviewer_ynom · 2025-10-23

**Soundness:** 2
**Presentation:** 3
**Contribution:** 2
**Rating:** 0
**Confidence:** 5

**Summary:**

The paper studies the phenomenon that negative prompts (NP) in CFG might push samples away from the negative conditions unboundedly, which severely harms the fidelity of the ground-truth conditional distribution and CFG with positive conditions. To this end, the authors proposes to reformulate the guidance weight through NCE loss. Such an adaptive methodology could alleviate the overemphasis of negative prompts in a milder and more flexible way.

**Strengths:**

- The design of adaptive guidance weights based on contrastive learning enhance the faithfulness of guided sampling both theoretically and experimentally.
- The whole pipeline is training-free and efficient, which needs almost no additional time cost.
- The experimental results confirms its efficacy compared with NP method.

**Weaknesses:**

- From theoretical perspective, contrastive learning forces the sample to appear similar to positive ones and different with negative ones, which is exactly the motivation of CFG from Bayesian theory using score functions and classification probabilities. The authors simply use a new probability function $p(x_{t-1}|x_t,c)$ instead of traditional $p(x_{t-1}|c)$ in CFG. By replacing the proposed function with what I raised, we have
  $$l^+=\log(1+\frac{p(x_{t-1}|\phi)}{p(x_{t-1}|c)})\approx\frac{p(x_{t-1}|\phi)}{p(x_{t-1}|c)}$$
  $$\nabla\frac{p(x_{t-1}|\phi)}{p(x_{t-1}|c)}=\frac{p(x_{t-1}|c)\nabla p(x_{t-1}|\phi)-p(x_{t-1}|\phi)\nabla p(x_{t-1}|c)}{p(x_{t-1}|c)^2}=\frac{p(x_{t-1}|c)p(x_{t-1}|\phi)(\nabla\log p(x_{t-1}|\phi)-\nabla\log p(x_{t-1}|c))}{p(x_{t-1}|c)^2}=\frac{p(x_{t-1}|\phi)}{p(x_{t-1}|c)}(\nabla\log p(x_{t-1}|\phi)-\nabla\log p(x_{t-1}|c))$$
  $$\epsilon_\phi-\gamma_t\nabla\frac{p(x_{t-1}|\phi)}{p(x_{t-1}|c)}=\epsilon_\phi+\gamma_t'(\nabla\log p(x_{t-1}|c)-\nabla\log p(x_{t-1}|\phi)),\quad\gamma_t'=\gamma_t\frac{p(x_{t-1}|\phi)}{p(x_{t-1}|c)}$$
This is almost the traditional CFG with some scaling term. Derivation of $l^-$ is similar. **So there is nothing new to involve contrastive learning than native CFG with Bayesian**. This is still correct without the first approximation, since we have
  $$\nabla\log(1+\frac{p(x_{t-1}|\phi)}{p(x_{t-1}|c)})=\frac{p(x_{t-1}|c)}{p(x_{t-1}|c)+p(x_{t-1}|\phi)}\nabla\frac{p(x_{t-1}|\phi)}{p(x_{t-1}|c)}$$
  $$\epsilon_\phi-\gamma_t\nabla l^+=\epsilon_\phi+\gamma_t''(\nabla\log p(x_{t-1}|c)-\nabla\log p(x_{t-1}|\phi)),\quad\gamma_t''=\gamma_t\frac{p(x_{t-1}|\phi)}{p(x_{t-1}|c)+p(x_{t-1}|\phi)}$$
- In spite of introducing contrastive learning regime in guidance weights, the proposed method is technically some **linear scaling** trick instead of previous fixed and identical guidance weights which are empirically set, which is of poor contribution. To be more detailed, native CFG use a fixed $\gamma$ for positive conditions and $-\gamma$ for negative ones, as is claimed in Eq. (9). What the authors propose is to use a new coefficient larger than 1 multiplying the positive weights ($\frac{2}{1+\exp{(-\tau\|\epsilon_{\phi}-\epsilon_c\|)}}>1$) and a new one smaller than 1 multiplying the negative weights ($\frac{2\exp{(-\tau\|\epsilon_{\phi}-\epsilon_c\|)}}{1+\exp{(-\tau\|\epsilon_{\phi}-\epsilon_c\|)}}<1$). This strengthens the positive signal and weakens the negative one, somewhat provides a more compact bound for negative guidance.
- What further confirms my opinion is Fig. 3. The proposed positive weights almost coincide with the native fixed ones, and the negative weights remain extremely small (smaller than 1). Therefore the proposed method is almost equivalent to introducing no negative prompts in CFG, considering that traditional CFG in text-to-image generation involves guidance weight greater than 5.
- The experiments are unfair and marginal. First, no significant improvements are addressed in Fig. 4, Tabs. 1-4. Second, no native CFG results are provided in Fig. 4-5, Tab. 2, and Tab. 4 (CLIP Score metric is missing). Third, no FID or FD_DINOv2 is reported on ImageNet but only classification.

**Questions:**

Please carefully discuss the relation between the proposed method and what I raised in Weaknesses, especially the behavior of traditional CFG with scaled weights. Besides, please add all missed quantitative results including CLIP Score, FID, and FD_DINOv2.

---

> ### Author Response · Authors · 2025-11-22
> **Rebuttal from Authors**
>
> We thank the reviewer ``ynom`` for the review and the feedback, and we provide detailed responses to address your remaining concerns:
>
> - We'd like to insist that simply treating all methods that use the term $(\epsilon_c - \epsilon_{\varnothing})$ as having no difference to CFG would be problematic; the sampling distribution changes completely depending on how the scale of the CFG term is adjusted, enabling various research to address[1, 2, 3]. For instance, DNG[3] shows that appropriate dynamic scaling of $(\epsilon_c - \epsilon_{\varnothing})$ transforms the unbounded negated CFG(=NP in our paper) sampling distribution of $p’(x) \propto p(x)p(c|x)^{-1}$ into the much more stable $p’(x) \propto p(x)(1-p(c|x))$. And we discussed how using negated constant-scale CFG terms for concept negation causes several problems in terms of their sampling distribution in Section 3.1.
> - Regarding the comments that the reviewer raised, we've discussed problems that arise when using a fixed scale in CFG with negative guidance(Section 3.1), and how CCFG can be seen as introducing an adaptive scale factor to the original CFG term(line 297), and this scale factor goes to 0 in negative guidance for sufficiently unrelated sample $x_t$(line 300), and how this can resolve the presented drawback of fixed-scale negated CFG(line 305), highlighting them as key contributions of our work.
> - Although the reviewer questions the idea of introducing contrastive loss to construct CFG, the reviewer's derivation also utilizes contrastive NCE loss and reached the same conclusion as ours; adaptive weights must be applied depending on how well $x_t$ aligns with the condition during sampling. And as we insisted above, this weight can dramatically alter the sampling distribution towards more accurate but stable outputs. Also, we defined the loss $\ell^+(\cdot)$ as a function of the denoising direction $\epsilon$ (Eq. 13), and computed the gradient of this loss w.r.t. the input $\epsilon$. This is the reason that we can add this gradient to the initial unconditional noise $\epsilon_{\varnothing}$ to obtain a new denoising direction in the spirit of ''optimization”. Meanwhile, the derivation suggested by the reviewer calculates all gradients with respect to the data domain $x_{t-1}$. Thus, we think adding this gradient to $\epsilon_{\varnothing}$ cannot be justified.
> - We'd like to note that we had mentioned in line 157\~162 that “NP” refers to “a (fixed scale) traditional CFG with a negated sign for negative guidance”. Therefore, all the experiments we’ve done, including Figures 4 and 5, already provide the performance of a traditional CFG. Also, the CLIPscore column is already in Tables 2 and 4.
> - To address the reviewer’s suggestion of distribution-alignment metrics(e.g., FID, FD-DINOv2) on the ImageNet1k task, we have extended our ImageNet-1k experiments to include the FD-DINOv2 metric[4]. Please refer to the revised Figure 4, where the experiments are done again with another constructed class pairs from a more systematic pipeline(revised Appendix D.2). CCFG outperforms the prevalent baselines in terms of both positive \& negative prompt alignment and the overall image quality(Aesthetic Score, FD-DINOv2).
>
> | (w=1.5) | **Pos Acc.** | **Neg. Acc.** | **Overall. Acc.** | **Aesthetic** | **FD-DINOv2** |
> | --- | --- | --- | --- | --- | --- |
> | CFG | 0.4959 | 0.9996 | 0.4957 | 4.0063 | 456.78 |
> | DNG | 0.8307 | 0.9886 | 0.8212 | 4.1115 | 215.07 |
> | PERP-NEG | 0.8544 | 0.9958 | 0.8508 | 4.2089 | 214.93 |
> | SLD | 0.7785 | 0.9945 | 0.7742 | 4.0943 | 246.19 |
> | **CCFG**($\tau_t$=0.5) | 0.9101 | 0.9932 | **0.9039** | **4.2535** | **190.71** |
>
> ---
>
> [1] Safe Latent Diffusion: Mitigating Inappropriate Degeneration in Diffusion Models, CVPR 2023
>
> [2] Applying Guidance in a Limited Interval Improves Sample and Distribution Quality in Diffusion Models, NeurIPS 2024
>
> [3] Dynamic Negative Guidance of Diffusion Models, ICLR 2025
>
> [4] Exposing flaws of generative model evaluation metrics and their unfair treatment of diffusion models, *NeurIPS 2023*.

---

> > ### Comment · Reviewer_ynom · 2025-11-22
> >
> > 1. I did not claim "use the term $(\epsilon_c - \epsilon_{\varnothing})$ as having no difference to CFG would be problematic", what I concerned is that **the proposed pipeline motivated by so-called contrastive learning has no novelty at all but a different perspective to explain CFG**. The only contribution is the adaptive guidance weight, so **the paper overclaims a lot**. The proposed method alleviates some issues in native CFG by **adaptive weights but not contrastive learning**.
> >
> > 2. Similar to the first point. If adaptive weight is the key contributions, then avoid to use any contrastive learning in the paper. Besides, adaptive weight in CFG is not new [1], which further weakens the contribution.
> >   [1] Classifier-free Guidance with Adaptive Scaling. Malarz et al., 2025.
> >
> > 3. Contrastive learning and Beyas theory are exactly the same. I use contrastive loss to prove **CFG with adaptive weights can also be derived by contrastive learning**, so the paper **has nothing new at all**. Besides, totally disagree about the $x_{t-1}$ words, the authors talk about the transition probability, *i.e.*, $p(x_{t-1}|x_t,c)$, which is some **approximate of the ground-truth one**, *i.e.*, $q(x_{t-1}|x_t)$. What I use, *i.e.*, $p(x_{t-1}|,c)$ can also be referred to as **the approximation of $p(x_t,c)$**.
> >
> > 4. Clarification of Fig. 3 is not convincing at all. The authors fail to address my concerns.
> >
> > 5. The quantitative results are poorly convincing. First, no FID is provided. Second, even EDM2-S could reach ~100 FD_DINOv2 on ImageNet 512 with no CFG. The reported results are confusing.

---

> > > ### Author Response · Authors · 2025-11-27
> > >
> > > We’d like to recall our main contribution described in our manuscript:
> > >
> > > - "CFG with fixed scale" is problematic for the negative guidance, because the guidance term remains significant even when the sample $x_t$ is already irrelevant to the negative prompt.
> > > - “Optimization of denoising direction $\epsilon$ to minimize the contrastive loss” leads us to derive "CFG with adaptive scale $\lambda$", where $\lambda$ can approach 0 when the negative prompt is sufficiently removed. This behavior of CCFG can resolve the issue in the negative CFG with a fixed scale.
> > >
> > > According to the above, we provide the response for the five comments from the reviewer:
> > >
> > > 1. Our adaptive weight scale is the direct consequence of utilizing contrastive loss from Eq.(12)\~Eq.(21), and these two cannot be separated; without introducing the NCE loss in Eq. (12), the formula for $\lambda$ in Eq.(18)&(21) cannot be derived. Also, "a fixed-scale CFG" cannot be inducted from the contrastive loss.
> > > 2. As we mentioned in the first response, we are aware that there are several works to apply adaptive weight into $(\epsilon_c - \epsilon_{\phi})$ term, but the key feature that characterizes these works is the motivation, derivation, behavior, and the effect of that adaptive weight.
> > > Apart from these previous works[1\~4], CCFG improves naive CFG on a broader task coverage of both positive and negative guidance, toward the sample distribution that is both stable and prompt-aligned(line 307~310).
> > > 3.
> > >     - Again, the reviewer’s derivation with contrastive loss and the conclusion of "CFG with adaptive weights can be derived by contrastive learning" coincides with what we've done in the paper. And unless this idea has already been presented in existing papers, one cannot simply claim it's not new. To refute the novelty of our work, the references that cover another adaptively-scaled CFG derived with contrastive loss have to be provided.
> > >     - In our first response, we pointed out the difference that we've utilized a denoising direction optimization via $\epsilon' = \epsilon_{\phi} - \gamma_t\nabla_{\epsilon}\ell(\epsilon_{\phi})$ (Eq. (17)), while the review apprently used $\epsilon' = \epsilon_{\phi} - \gamma_t\nabla_{x_{t-1}}\ell(x_{t-1})$ in the derivation (not about the choice between $p(x_{t-1}|c)$ or $p(x_{t-1}|x_t, c)$).
> > > 4. About Figure 3, the behavior of a guidance term that the reviewer described in the primary review is identical to lines 298~305. Moreover, as we stated in the recall above, this is the "benefit" of CCFG that addresses the drawback of naive fixed-scale CFG, rather than a "concern".
> > > 5. For the mentioned FD_DINOv2, there are two factors that can make the FD_DINOv2 differ from the previously known value: first, our experiment is different from a conventional unconditional/conditional generation, but sampling with both positive class and negative class (that are perceptually similar to the positive class to hinder the positive guidance) conditions. Second, since only a subset of all 1,000 classes is utilized, the reference distribution was changed accordingly to only contain the classes that we've used as a positive class.
> > >
> > > ---
> > >
> > > [1] Safe Latent Diffusion: Mitigating Inappropriate Degeneration in Diffusion Models, CVPR 2023
> > >
> > > [2] Applying Guidance in a Limited Interval Improves Sample and Distribution Quality in Diffusion Models, NeurIPS 2024
> > >
> > > [3] Dynamic Negative Guidance of Diffusion Models, ICLR 2025
> > >
> > > [4] Classifier-free Guidance with Adaptive Scaling. arxiv, 2025.

---

> > > > ### Comment · Reviewer_ynom · 2025-11-27
> > > >
> > > > I insist that the whole paper is weak at both technical and theoretical aspects. Using a similar theory (*i.e.*, contrastive learning) to explain some widespread technique cannot be referred as novelty. Again, I reaffirm that the paper overclaims a lot, since it has nothing to do with contrastive learning. In a more detailed way, the authors first point out **negative prompt is problematic**, then **contrastive loss helps facilitate this issue**. However, **only contrastive loss cannot facilitate at all** since it is theoretically equivalent to CFG. So the whole paper is with severe logic issue.
> > > >
> > > > Given that the authors fail to understand my problems and the experiments are not convincing, I maintain my score with strong belief.

---

> > > > > ### Author Response · Authors · 2025-11-28
> > > > >
> > > > > Despite our clarifications in two earlier responses, the reviewer continues to claim that “contrastive learning is equivalent to CFG.” We must strongly object: this is a clear reflection of a serious lack of familiarity with the fields of CFG and contrastive (NCE) learning.
> > > > >
> > > > > A scientifically meaningful critique should provide rigorous theoretical grounds and/or credible references, rather than rely on unsupported assertions. We caution against including unfounded claims in a formal review, as this risks misinforming readers and undermining the integrity of the evaluation process.
> > > > >
> > > > > The difference between CFG and contrastive learning is fundamental and should be crystal clear even at an introductory level. To clarify these basic points for the reviewer, we briefly recall the elementary “101” definitions of CFG and contrastive learning.
> > > > >
> > > > > CFG [1] is designed to sample from the sharpened distribution
> > > > > **$p’(x) \propto p(x)p(c|x)^{w}$**,
> > > > > whereas contrastive learning in the sense of NCE [2] relies on logistic regression between data–noise pair distributions. In particular, contrastive learning explicitly requires comparisons between multiple samples (data vs. noise), whereas CFG is derived from a single-sample conditional density. These are fundamentally different constructions, and this distinction should be evident to any reader with basic training in probabilistic modeling. Any claim of equivalence between the two therefore requires strong theoretical justification and appropriate references, which the current review does not provide.
> > > > >
> > > > > ---
> > > > >
> > > > > [1] Classifier-free Diffusion Guidance, NeurIPS 2021 Workshop
> > > > >
> > > > > [2] Noise-contrastive estimation: A new estimation principle for unnormalized statistical models, AISTATS 2010.

---

### Official Review · Reviewer_1rja · 2025-10-31

**Soundness:** 2
**Presentation:** 3
**Contribution:** 2
**Rating:** 4
**Confidence:** 3

**Summary:**

This paper introduces Contrastive Classifier-Free Guidance (CCFG), a novel diffusion sampling mechanism that reformulates conditional guidance as an optimization problem based on a contrastive loss framework. CCFG successfully addresses the limitations of traditional Negative Prompting (NP), which suffers from sample distortion and quality degradation due to unbounded inverse probability distributions. The core innovation is the negative guidance term whose weight automatically approaches zero when the sample is irrelevant to the negative concept, preventing unnecessary pushing into low-density areas and enhancing sampling stability. CCFG achieves superior performance in handling both positive and negative conditions. Experiments demonstrate that, in both class-conditional generation and text-to-image tasks, CCFG more reliably and thoroughly excludes unwanted concepts, improving alignment with positive prompts while maintaining perceptual sample quality.

**Strengths:**

1. This paper proposes CCFG, which offers a theoretically sound mechanism by reformulating conditional guidance using a contrastive loss framework.
2. The proposed CCFG resolves the critical issue of distribution distortion and sample quality degradation inherent in standard Negative Prompting.
3. Quantitative and qualitative experiments show the proposed method’s effectiveness.

**Weaknesses:**

1.	The paper lacks a detailed sensitivity analysis on the crucial hyper-parameters τ_t and ω_t, hindering reproducibility and generalizability.
2.	The authors should compare the proposed CCFG with more recent Classifier-Free Guidance methods, such as CFG++[1], Autoguidance[2], Safree[3] on SD1.5/SDXL and also CFG-zero[4] on flow-matching model, for a fairer assessment.
3.	The advantages of CCFG over comparing methods on quantitative experiments are relatively modest. From Fig.4 and Tab.2, the CCFG’s scores are often only slightly higher or sometimes lower than other methods, raising the question about whether the theoretical analysis lead to substantial and stable improvement.
4.	The qualitative comparisons in Fig.5 is difficult to clearly see the advantages of CCFG over other methods. It is recommended to highlight the differences for better readability.
5.	The paper does not provide a comparison of the inference time or computational overhead introduced by CCFG compared with standard CFG or other complex guidance methods.

[1] Chung H, Kim J, Park G Y, et al. Cfg++: Manifold-constrained classifier free guidance for diffusion models[J]. arXiv preprint arXiv:2406.08070, 2024.

[2] Karras T, Aittala M, Kynkäänniemi T, et al. Guiding a diffusion model with a bad version of itself[J]. Advances in Neural Information Processing Systems, 2024, 37: 52996-53021.

[3] Yoon J, Yu S, Patil V, et al. Safree: Training-free and adaptive guard for safe text-to-image and video generation[J]. arXiv preprint arXiv:2410.12761, 2024.

[4] Fan W, Zheng A Y, Yeh R A, et al. Cfg-zero*: Improved classifier-free guidance for flow matching models[J]. arXiv preprint arXiv:2503.18886, 2025.

**Questions:**

Please refer to the questions and suggestions in the “Weaknesses” part.

---

> ### Author Response · Authors · 2025-11-22
> **Rebuttal from Authors**
>
> We thank Reviewer ``1rja`` for the constructive feedback. Below, we provide point-by-point responses on your questions and concerns:
>
> - W1. The paper lacks a detailed sensitivity analysis on the crucial hyper-parameters τ_t and ω_t, hindering reproducibility and generalizability.
>
> We appreciate the request for a more systematic characterization of the hyperparameters. Please refer to Figure 7, where we show the performance of CCFG on SD1.5 with PIE-Bench-Neg with different selections of $\omega_t$ and $\tau_t$.
>
> Regarding $\omega_t$, we emphasize that it plays a role analogous to the guidance scale $\omega_t$ in a traditional CFG. The positive prompt alignment($VLM_{pos}$) and negative prompt removal($VLM_{neg}$) improve as the guidance scale $\omega_t$ increases, while the overall quality of the image(Aesthetic Score) has a sweet spot at a suitable $\omega_t$. This trend agrees with a well-known observation of traditional CFG.
> Meanwhile, as we have illustrated in Figure 6-(b), a smaller $\tau_t$ can more thoroughly remove the negative concept, but also hinders the positive prompt alignment. As $\tau_t$ reaches 0, CCFG becomes equivalent to the traditional CFG.
>
> - W2. The authors should compare the proposed CCFG with more recent Classifier-Free Guidance methods, such as CFG++[1], Autoguidance[2], Safree[3] on SD1.5/SDXL and also CFG-zero[4] on flow-matching model, for a fairer assessment.
>
> As the reviewer suggested, we have expanded our baselines as follows. For the positive-only guidance scenario, we add CFG++  for the diffusion models (SD1.5/SDXL) and CFG-zero for the flow-matching model (SD3). For the positive & negative guidance scenario, we add SAFREE as the extra baseline. Please note we do not include Autoguidance as a baseline, as it requires the construction of a “bad model” and there is currently no well-established standard for preparing such a model in the SD1.5/SDXL or SD3 settings.
>
> CCFG continues to achieve the highest overall performance across the evaluated settings. In particular, CCFG either matches or improves upon CFG++ and CFG-zero in a positive-only guidance scenario, and consistently outperforms SAFREE in the positive+negative guidance scenario.
>
> | GenEval | Method | single_obj. | two_obj. | count | colors | position | color_attr. | overall |
> | --- | --- | --- | --- | --- | --- | --- | --- | --- |
> | SD1.5 | CFG++ | **0.991** | **0.434** | **0.290** | 0.739 | 0.047 | 0.055 | 0.426 |
> |  | CCFG(ours) | 0.986 | 0.422 | 0.284 | **0.782** | **0.058** | **0.058** | **0.432** |
> | SDXL | CFG++ | 0.972 | 0.691 | **0.453** | 0.872 | **0.128** | 0.225 | 0.556 |
> |  | CCFG(ours) | **0.984** | **0.717** | 0.441 | **0.886** | 0.123 | **0.240** | **0.565** |
> | SD3 | CFG-zero | 0.984 | **0.765** | 0.550 | 0.816 | **0.257** | **0.482** | 0.642 |
> |  | CCFG(ours) | **0.988** | 0.748 | **0.575** | **0.840** | 0.250 | 0.463 | **0.644** |
>
> |  | Positive prompt[↑] |  |  |  | Negative prompt[↓] |  |  |  | Overall[↑] |  |
> | --- | --- | --- | --- | --- | --- | --- | --- | --- | --- | --- |
> | **SD1.5** | CLIP | IR | HPSv2 | VLM | CLIP | IR | HPSv2 | VLM | VLM$_{all}$ | Aesthetic |
> | SAFREE | 0.2658 | 0.4329 | 0.2595 | 0.8272 | 0.1341 | **-1.9551** | **0.1709** | 0.0909 | 0.7470 | 5.5969 |
> | CCFG(ours) | **0.2703** | **0.5791** | **0.2695** | **0.8540** | **0.1339** | -1.9356 | 0.1709 | **0.0894** | **0.7777** | **5.6211** |
> | **SD3** | CLIP | IR | HPSv2 | VLM | CLIP | IR | HPSv2 | VLM | VLM$_{all}$ | Aesthetic |
> | SAFREE | 0.2729 | 1.1831 | 0.2962 | 0.9160 | 0.1389 | **-1.8529** | **0.1781** | 0.1182 | 0.8077 | 5.9987 |
> | CCFG(ours) | **0.2739** | **1.1840** | **0.2978** | **0.9164** | **0.1381** | -1.8234 | 0.1790 | **0.1139** | **0.8120** | **6.0158** |

---

> > ### Author Response · Authors · 2025-11-22
> > **Rebuttal from Authors (continue)**
> >
> > - W3. The advantages of CCFG over comparing methods on quantitative experiments are relatively modest. From Fig.4 and Tab.2, the CCFG’s scores are often only slightly higher or sometimes lower than other methods, raising the question about whether the theoretical analysis lead to substantial and stable improvement.
> >
> > We appreciate the reviewer’s comment. Please refer to the revised Figure 4, where the class pairs were constructed in a more systematic pipeline(revised Appendix D.2), and CCFG outperforms the prevalent baselines NP, DNG, Perp-Neg, and SLD across *all* class-conditioned settings. On MNIST and CIFAR-10, CCFG achieves superior FID and higher Negation ratios compared to NP and DNG. On ImageNet-1k, CCFG achieves the best FD-DINOv2, Aesthetic score, and overall classification accuracy, indicating both high perceptual quality and strong alignment with the target classes.
> >
> > - W4. The qualitative comparisons in Fig.5 is difficult to clearly see the advantages of CCFG over other methods. It is recommended to highlight the differences for better readability.
> >
> > Following the reviewer’s helpful suggestion, we have revised Figure 5 to explicitly highlight the differences. We marked the regions in the examples where either negative prompt features have not been fully removed, or where the positive prompt has not been faithfully followed.
> >
> > - W5. The paper does not provide a comparison of the inference time or computational overhead introduced by CCFG compared with standard CFG or other complex guidance methods.
> >
> > CCFG uses the same number of model forward passes as standard CFG or other prevalent methods (such as DNG, SLD, and Perp-Neg). The only additional computation in CCFG is the calculation of $\lambda^+$ and $\lambda^-$, whose cost is almost negligible compared to the actual model forward pass. Therefore, we ensure that there is no practical computational overhead gain compared to naive CFG or other complex guidance methods in terms of the wall-clock time or VRAM.

---

### Official Review · Reviewer_V8UV · 2025-11-01

**Soundness:** 3
**Presentation:** 2
**Contribution:** 4
**Rating:** 6
**Confidence:** 4

**Summary:**

The paper proposes Contrastive CFG, a guidance method for diffusion sampling that handles positive and negative concepts via an NCE perspective. It targets a known issue with “negative prompts” implemented as naïvely negated CFG terms, which can invert the distribution and degrade image quality. The method is claimed to both inject and remove specified concepts while preserving fidelity, across simple class labels and more complex text prompts.

**Strengths:**

1. **Conceptually novel and well-motivated.** The paper connects the behavior of negative prompts to an NCE objective and proposes a principled fix, achieving strong reported results.
2. **Broad and diverse evaluation.** Beyond standard class-conditional generation, the authors introduce a more realistic benchmark with contrastive prompts.
3. **Clear intuition.** Figure 2(b) and its accompanying experiment are intuitive and effectively illustrate the core idea.

**Weaknesses:**

1. **Generalization concerns**
   - **Sampler coverage.** The paper focuses on DDIM (deterministic). Please evaluate a stochastic sampler (e.g., DDPM). Since the method acts directly on the sampling process, stochasticity might reveal interesting behavior.
   - **Backbone diversity.** Can the approach generalize to ViT-based diffusion backbones? In paper, authors only consider SD1.x and SDXL.

2. **Missing detail in benchmarks and setup**
   - **Class-pair construction.** Please elaborate on how the 100 class pairs with strong visual proximity are selected, and include more examples in the appendix.
   - **PIE-Bench-Neg clarity.** The construction is currently vague. Provide concrete prompt examples.
   - **Sampling/config details (Tables 1 &3 & 4).** Please specify how CFG and CCFG are configured (e.g., guidance scales, sampling steps, schedulers) to enable exact reproducibility.

3. **Limited ablation**
   - **Guidance strength.** Include ablations across guidance weights (and any temperature/margin in the contrastive objective) to characterize stability and trade-offs.

**Questions:**

1. **About PIE-Bench-Neg.** The appendix mentions four categories. Can you report per-category results to reveal where the method helps most. Also analyze whether negative prompts being **adjectives** (style/attribute changes) versus **nouns** (object removal) affects performance.  It is very interesting.
2. **Failure cases.** Can you visualize typical failure modes and give some explanation?

Overall, the paper is **novel and effective**, with a compelling motivation and promising results. However, several points remain underspecified. If you can address my concerns, I will consider raising my score.

---

> ### Author Response · Authors · 2025-11-22
> **Rebuttal from Authors**
>
> Thank you for the reviewer ``V8UV``'s thoughtful feedback and for considering our work, and we appreciate your time and evaluation. Below, we provide point-by-point responses on your questions and concerns.
>
> - **W1. Generalization concerns**
>     - **Sampler coverage.** The paper focuses on DDIM (deterministic). Please evaluate a stochastic sampler (e.g., DDPM). Since the method acts directly on the sampling process, stochasticity might reveal interesting behavior.
>
>     As the reviewer suggested, we introduced Markovian stochasticity in the sampling process and found that the overall performance of CCFG remains almost unchanged compared to the deterministic sampling. For both positive and negative prompts, both sampling processes achieve similar performance arcross the CLIP, ImageReward, HPSv2 and VLM score. Please refer to the Table below for the results on PIE-Bench-Neg with SD1.5:
>
>     |  | Positive prompt |  |  |  | Negative prompt |  |  |  |
>     | --- | --- | --- | --- | --- | --- | --- | --- | --- |
>     |  | CLIP | IR | HPSv2 | VLM | CLIP | IR | HPSv2 | VLM |
>     | Markovian | 0.2710 | 0.5851 | 0.2653 | 0.8536 | 0.1341 | -1.9387 | 0.1711 | 0.0901 |
>     | Deterministic | 0.2703 | 0.5791 | 0.2695 | 0.8540 | 0.1339 | -1.9356 | 0.1709 | 0.0894 |
>
>     - **Backbone diversity.** Can the approach generalize to ViT-based diffusion backbones? In paper, authors only consider SD1.x and SDXL.
>
>     We’d like to kindly remind the reviewer that we utilized ImageNet-1k pre-trained DiT-XL/2 for our experiments on ImageNet-1k and used SD3(w/ MMDiT backbone) for experiments on flow-matching models. For both cases, CCFG consistently improves alignment with positive prompts while effectively suppressing negative prompts, thereby demonstrating the generalizability of CCFG to ViT-based diffusion backbones.
>
>
> - **W2. Missing detail in benchmarks and setup**
>     - **Class-pair construction.** Please elaborate on how the 100 class pairs with strong visual proximity are selected, and include more examples in the appendix.
>
>     We thank the reviewer for pointing out the need for clarification. We primarily utilized the ImageNet hierarchy information between the classes given by WordNet[1]. In the original submission, the 100 class pairs were manually curated based on the closely located class nodes. However, for the reproducibility in the revised paper, we replace this with a fully programmatic pipeline:
>
>     1. For each ImageNet-1k class, we retrieve the corresponding WordNet ID (wnid) and its hypernyms.
>     2. We construct a pool of candidate pairs by including all pairs of classes that share an identical hypernym.
>     3. From this pool, we uniformly sample 100 class pairs to obtain the final benchmark set, while ensuring that every class in the 100 class pairs is unique.
>
>     Please refer to Appendix D.2, where we include this updated pipeline and several examples from the 100 pairs used in the experiment.
>
>     - **PIE-Bench-Neg clarity.** The construction is currently vague. Provide concrete prompt examples.
>
>     Following the reviewer’s suggestion, we have revised Appendix D.3 to provide a clearer and more concrete description of PIE-Bench-Neg. The updated appendix now includes explicit examples for each category of negative prompts, illustrating how positive and negative conditions are constructed and how they differ (e.g. object removal, attribute negation, style changes).
>
>     - **Sampling/config details (Tables 1 &3 & 4).** Please specify how CFG and CCFG are configured (e.g., guidance scales, sampling steps, schedulers) to enable exact reproducibility.
>
>     For SD15(Table 1), we sampled with NFE=50 deterministic DDIM sampling with uniform timestep scheduling, using w=7.5 for both CFG and CCFG. In SD3(Table 3 & 4), the NFE and the guidance scale w are changed to NFE=28 and w=3.0, respectively. Please refer to Appendix D.3. The $\tau$ value for CCFG on each model is listed in Table 5.
>
>
> - **W3. Limited ablation**
>     - **Guidance strength.** Include ablations across guidance weights (and any temperature/margin in the contrastive objective) to characterize stability and trade-offs.
>
>     We appreciate the request for a more systematic characterization of the hyperparameters. In the revised version, we include an ablation study (Figure 7) on SD1.5 with PIE-Bench-Neg, with different $\omega_t$ and $\tau_t$.
>
>     The positive prompt alignment($VLM_{pos}$) and negative prompt removal($VLM_{neg}$) improve as the guidance scale $\omega_t$ increases, while the overall quality of the image(Aesthetic Score) has a sweet spot at a suitable $\omega_t$. This trend agrees with a well-known observation of traditional CFG. Meanwhile, as we have illustrated in Figure 6-(b), a smaller $\tau_t$ can more thoroughly remove the negative concept, but also hinders the positive prompt alignment. As $\tau_t$ reaches 0, CCFG becomes equivalent to the traditional CFG.

---

> > ### Author Response · Authors · 2025-11-22
> > **Rebuttal from Authors (continue)**
> >
> > - **Q1. About PIE-Bench-Neg.** The appendix mentions four categories. Can you report per-category results to reveal where the method helps most. Also analyze whether negative prompts being **adjectives** (style/attribute changes) versus **nouns** (object removal) affects performance. It is very interesting.
> >
> > We segmented the positive/negative prompt pairs in PIE-Bench-Neg into nine subgroups, according to the category that the original prompt in PIE-Bench belongs to. The performance of CCFG in each of these subgroups is shown in Figure 10 in Appendix D.3.
> >
> > Here, we observed that CCFG is relatively less effective on the series of “*change_attribute_XX*” categories, which tend to yield relatively lower positive prompt VLM scores(*e.g.*, change_attr_color) or higher negative prompt VLM scores(*e.g.*, change_attr_pose). This can be because the negative prompts from these categories require removing specific attributes of something explicitly contained in the positive prompt, therefore conveying overlapping concepts and making the task more difficult. We also found that having nouns as negative prompts(*e.g.*, change_object) or adjectives(*e.g.*, change_style) shows similar performances.
> >
> > - **Q2. Failure cases.** Can you visualize typical failure modes and give some explanation?
> >
> > Following the reviewer’s suggestion, we report several failure cases in Figure 9 of Appendix B.5; when complex negative prompts such as multi-attribute prompts are used, the method may fail to remove all of the specified attributes.  And when the undesired negative prompt describes subtle continuous attributes, the negative guidance strength of CCFG may be weakened in the process of $x_t$ being repelled from the unwanted concept, resulting in outputs on the borderline of unwanted features.
> >
> > ---
> >
> > [1] ImageNet hierarchy, [https://observablehq.com/@mbostock/imagenet-hierarchy](https://observablehq.com/@mbostock/imagenet-hierarchy)

---

### Official Review · Reviewer_xnMb · 2025-11-01

**Soundness:** 2
**Presentation:** 2
**Contribution:** 2
**Rating:** 4
**Confidence:** 3

**Summary:**

This paper proposes a contrastive guidance method for conditional diffusion models that improves upon traditional Classifier-Free Guidance (CFG) and Negative Prompting (NP). The method uses a contrastive loss to align or repel the denoising direction based on the given condition. This approach achieves effective concept injection or removal while preserving image quality across various conditions, from simple classes to complex text prompts.

**Strengths:**

- This paper proposes a framework for designing the coefficients of positive and negative guidance in CFG.
- The method shows only marginal improvement.

**Weaknesses:**

* The method needs one more forward times compared with classic positive minus negative way.

*  Still lack the analytic closed-form sampling distribution.

**Questions:**

- Can the framework provide design choices or strategies for scenarios where only two inference steps are allowed for each step?

---

> ### Author Response · Authors · 2025-11-22
>
> We appreciate the reviewer ``xnMb`` for your detailed comments and valuable suggestions. Below, we provide thorough point-by-point responses to address your concerns.
>
>
> - W1: The method needs one more forward times compared with classic positive minus negative way.
>
> While CCFG does require the diffusion model pass for the unconditional, positive, and negative conditions at each sampling step, we would like to emphasize that this is not a unique drawback of CCFG, but rather a common requirement for methods that simultaneously handle both positive and negative prompts. In particular, the baselines we compare against(Perp-Neg, SLD, DNG, etc.) and other negative-prompting variants[1] also require all three noise predictions to construct their guidance terms. Therefore, as CCFG is used in the same experimental setting as these prevalent methods, we consider CCFG not to require specifically costly computation.
>
>
> - W2: Still lack the analytic closed-form sampling distribution.
>
> We agree that we do not provide a closed-form expression for the sampling distribution $p_t(x)$ induced by CCFG, as we have clarified this point in the conclusion. That said, we would like to emphasize that while the explicit sampling distribution formulation might give us some insight into the sampling behavior, the primary objective of CCFG is to improve practical guidance behavior(i.e., reliably injecting or removing given concepts). In the same spirit as other widely adopted guidance methods[2, 3], we believe that the lack of an analytic closed-form sampling distribution does not fundamentally weaken our contribution.
>
> - Q1: Can the framework provide design choices or strategies for scenarios where only two inference steps are allowed for each step?
>
> Since CCFG is designed to “separately and adaptively” control positive and negative guidance via a contrastive loss between three noise predictions of $\epsilon_{\phi}, \epsilon_{c^+}, \epsilon_{c^-}$, three model passes are required under the current construction of CCFG. This is precisely what enables CCFG to balance positive and negative prompts without over-repelling or under-repelling, as demonstrated in our experiments. And as we insisted in the response of W1, this reliance on three evaluations is not only specific to our method, but also to many works that aim to address positive and negative conditions.
>
> ---
>
> [1] SAFREE: Training-Free and Adaptive Guard for Safe Text-to-Image And Video Generation, ICLR 2025
>
> [2] Applying Guidance in a Limited Interval Improves Sample and Distribution Quality in Diffusion Models, NeurIPS 2024
>
> [3] Guiding a Diffusion Model with a Bad Version of Itself, NeurIPS 2024

---

### Meta-Review · Area_Chair_jva8 · 2025-12-15

**Summary:**

The paper aims at addressing the problem of negative prompts in conditional diffusion models by introducing a contrastive CFG with a positive or negative prompt-conditioned contrastive loss.

Strengths identified by reviewers include, its efficiency in computational cost, well-motivated and broadly evaluated (Reviewer V8UV) and its focus on resolving a practical problem.

However, the paper also faces several critical weaknesses that have not been addressed in the rebuttal. Reviewer ynom raised the major concerns that the idea of contrasting positive and negative examples are already embedded in the classic CFG and estimating adaptive weight in CFG is explored in a recent work. Other concerns include marginal improvement of the proposed method (reviewer xnMb, ynom, 1rja) and multiple forwards requires higher computational cost (xnMb, 1rja).


Given these considerations, this paper clearly fails to meet the standards of ICLR. The authors are encouraged to address the highlighted issues to strengthen their contribution to the field of Diffusion Models.

**Reviewer Concerns:**

The authors provided detailed feedback and most of concerns have been addressed including lack of analytic closed-form sampling distribution (xnMb), generalization concerns, missing detail in benchmarks and setup, and limited ablation study (xnMb), more detailed experiments on crucial hyper parameters, and comparisons with more recent CFG methods (1rja).

However, several concerns including clearly explain the relations between contrasting positive and negative examples and the classic CFG (ynom),  estimating adaptive weight in CFG is explored in a recent work (ynom), marginal improvement of the proposed method (reviewer xnMb, ynom, 1rja) and multiple forwards requires higher computational cost (xnMb, 1rja).

**Reviewer Scores:**

The initial reviewer scores are mixed (two borderline reject, one strong reject and one borderline accept). Reviewer ynom has confirmed keeping initial negative score (strong reject) and other reviewers would have maintained their scores.

---

### Decision · Program_Chairs · 2026-01-26

Reject